# Spatial transcriptomic and single-nucleus analysis reveals heterogeneity in a gigantic single-celled syncytium

Tobias Gerber[1†], Cristina Loureiro[2†], Nico Schramma[3†], Siyu Chen[3,4], Akanksha Jain[2], Anne Weber[3], Anne Weigert[1], Malgorzata Santel[2], Karen Alim[3,4*], Barbara Treutlein[1,2*], J Gray Camp[1,5,6*]

[1]Max Planck Institute for Evolutionary Anthropology, Leipzig, Germany; [2]Department of Biosystems Science and Engineering, ETH Zürich, Basel, Switzerland; [3]Max Planck Institute for Dynamics and Self-Organization, Göttingen, Germany; [4]Physics Department, Technical University of Munich, München, Germany; [5]Roche Institute for Translational Bioengineering (ITB), Roche Pharma Research and Early Development, Roche Innovation Center, Basel, Switzerland; [6]University of Basel, Basel, Switzerland

**\*For correspondence:**
k.alim@tum.de (KA);
barbara.treutlein@bsse.ethz.ch (BT);
jarrettgrayson.camp@unibas.ch (JGC)

[†]These authors contributed equally to this work

**Abstract** In multicellular organisms, the specification, coordination, and compartmentalization of cell types enable the formation of complex body plans. However, some eukaryotic protists such as slime molds generate diverse and complex structures while remaining in a multinucleate syncytial state. It is unknown if different regions of these giant syncytial cells have distinct transcriptional responses to environmental encounters and if nuclei within the cell diversify into heterogeneous states. Here, we performed spatial transcriptome analysis of the slime mold *Physarum polycephalum* in the plasmodium state under different environmental conditions and used single-nucleus RNA-sequencing to dissect gene expression heterogeneity among nuclei. Our data identifies transcriptome regionality in the organism that associates with proliferation, syncytial substructures, and localized environmental conditions. Further, we find that nuclei are heterogenous in their transcriptional profile and may process local signals within the plasmodium to coordinate cell growth, metabolism, and reproduction. To understand how nuclei variation within the syncytium compares to heterogeneity in single-nucleus cells, we analyzed states in single *Physarum* amoebal cells. We observed amoebal cell states at different stages of mitosis and meiosis, and identified cytokinetic features that are specific to nuclei divisions within the syncytium. Notably, we do not find evidence for predefined transcriptomic states in the amoebae that are observed in the syncytium. Our data shows that a single-celled slime mold can control its gene expression in a region-specific manner while lacking cellular compartmentalization and suggests that nuclei are mobile processors facilitating local specialized functions. More broadly, slime molds offer the extraordinary opportunity to explore how organisms can evolve regulatory mechanisms to divide labor, specialize, balance competition with cooperation, and perform other foundational principles that govern the logic of life.

## Editor's evaluation

Single-celled organisms are assumed to be smaller, simpler, and less complex than multicellular organisms like animals. Here, the authors provide evidence for variation in gene expression in the syncytial (multinucleate) large amoeba *Physarum polycephalum*. This study is an elegant and interesting regarding heterogeneity of gene expression patterns and thus specialization of functions within a syncytial organism.

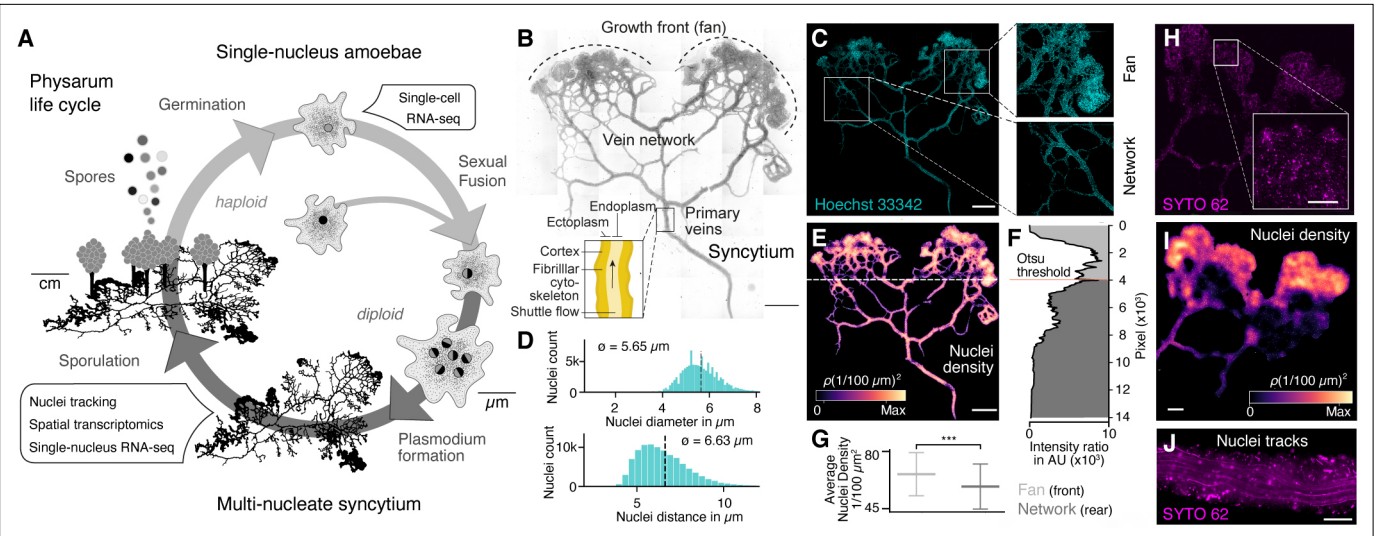

**Figure 1.** The slime mold *Physarum polycephalum* forms a multinucleate syncytium with mobile nuclei. (**A**) Simplified life cycle of *P. polycephalum* indicating which experiments were performed in this study. (**B**) Plasmodium of *P. polycephalum* grows as a hierarchical network composed of contractile veins and a microstructured growth front. Tiled brightfield image of a fixed plasmodium is shown as reference. Scale bar: 1000 μm. (**C**) Hoechst-stained nuclei across the syncytium from (**B**) a growth front (top right) and a primary vein (bottom right). Scale bar: 1000 μm. (**D**) Histograms of nuclei diameter (top) and density (bottom) estimates are shown. Mean values are provided as insets. (**E**) Density map of nuclei after StarDist segmentation and kernel density estimation shows inhomogeneously distributed nuclei (see Materials and methods for further details). Scale bar: 1000 μm. (**F**) Averaged relative density estimates across the x-axis of the plasmodium are visualized along the y-axis of the plasmodium. A threshold by *Otsu, 1979* was estimated to unbiasedly find a change in the density distribution separating the plasmodium in a network and a fan region. (**G**) Boxplots visualize the average nuclei density for the fan or network region identified in (**F**). p-value<0.001: *** (independent *t*-test, p=3.51 × 10⁻¹⁰). (**H**) Tiled life image of a plasmodium stained with SYTO 62 and a magnified fan region (inset). Scale bar: 500 μm. (**I**) Density map of nuclei shows that nuclei are not distributed homogeneously. Scale bar: 500 μm. (**J**) Maximum intensity projection over a period of 5 s of nuclei motion visualized by streamlines. See *Video 1*. Scale bar: 100 μm.

The online version of this article includes the following figure supplement(s) for figure 1:

**Figure supplement 1.** Nuclei are not homogeneously distributed across the multinucleate plasmodium.

## Introduction

Animals and other multicellular organisms are composed of diverse, compartmentalized cell types that perform specialized functions. Coordinated behavior between cells has led to the evolution of complex body plans with specialized organs. Furthermore, intercellular interactions between cell types are required to maintain a balanced physiology in a dynamic environment. The need for coordinated behavior over large spatial domains led in some cases to the formation of multinucleate cells, also called syncytia. Examples are skeletal muscles of animals (reviewed in *Abmayr and Pavlath, 2012*), the developing endosperm in many plants (coenocytes) (*Bennett et al., 1997*), multinucleate hyphae in different fungi (*Mela et al., 2020*), or the plasmodia of many protists (*Collins, 1969*; *Filosa and Dengler, 1972*; *Kerr, 1967*; *Guttes et al., 1961*). The syncytium, a cytoplasmic mass containing numerous nuclei that store genetic material including deoxyribonucleic acid (DNA), originates either through merging of cells or through nuclear divisions lacking an accompanying cell division. An example for the latter mode of syncytia formation are the plasmodia of many slime molds, such as *Physarum polycephalum*. Interestingly, such acellular slime molds are able to generate complex and dynamic body structures covering tens of centimeters or more while lacking compartmentalization of nuclei into discrete cellular units. Yet, nuclei divisions can occur through synchronized waves (*Howard, 1932*) or as independent (asynchronous) events (*Kerr, 1988*) across the noncompartmentalized cytoplasm. In addition, acellular slime molds are unique model organisms that can autonomously switch between a mitotic whole-cell division and a nuclei division giving rise to the plasmodial syncytium. However, the underlying molecular processes that lead to syncytium formation and enable different subdomains to respond to dynamic environments are poorly understood.

Here, we study *P. polycephalum*, a slime mold with a multi-phasic life cycle where uninucleate haploid amoeba merge to form a diploid zygote that grows into a gigantic protoplasmic syncytium,

called a plasmodium (*Figure 1A*; *Dove et al., 2012*). The plasmodium forms a network-like structure to span and connect various nutrient sources (*Boddy et al., 2009*; *Figure 1B*). The morphologically complex *Physarum* plasmodium can grow meters in area, exhibit diverse phenotypic behaviors, and dynamically respond to environmental conditions that it encounters (*Dussutour et al., 2010*). Striking examples of the diverse behavioral repertoire of this single-celled organism include that *Physarum* can solve the fastest way through a maze (*Nakagaki et al., 2000a*), anticipates periodic stimuli (*Saigusa et al., 2008*), or optimizes the cytoplasmic flow to more quickly escape unfavorable conditions (*Bäuerle et al., 2020*). Furthermore, the plasmodium's network-like body plan consists of interlaced tubes of varying diameters, and it was recently shown that these tubes grow and shrink

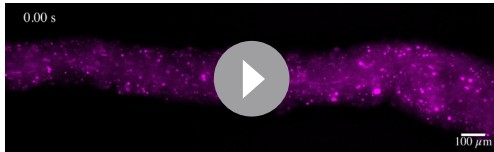

**Video 1.** Nuclei shuttle flow in a primary vein of a network region.
Live recording of SYTO62-stained nuclei in a primary vein of a network region of a *Physarum* plasmodium. Nuclei shuttle within the cytoplasmic flow. While some nuclei get trapped in the ectoplasm, others are released from the ectoplasm into the cytoplasmic flow. Differences in nuclei sizes originate from the single imaging plane of the three-dimensional tubes of the plasmodium and rarely by contamination of the autofluorescence of larger vesicles.
https://elifesciences.org/articles/69745/figures#video1

in diameter in response to a nutrient source, thereby functioning to imprint the nutrient's location in the tube diameter hierarchy (*Kramar and Alim, 2021*). The syncytium is composed of tens to millions of nuclei depending on the plasmodium size, and the nuclei are thought to divide synchronously as it grows (*Howard, 1932*; *Schiebel, 1973*). Studies have described that nuclei can be mobile and flow dynamically within cytoplasmic spaces, moving throughout the plasmodium (*Earnshaw and Steer, 1983*; *Boussard et al., 2021*) with up to 1.3 mm/s, which is among the fastest cytoplasmic flows ever measured (*S Mogre et al., 2020*; *Kamiya, 1950*). It is not known if nuclei diversify into heterogeneous states, if nuclei divisions are controlled locally or systemically, if there are gene expression modules that guide the formation of the complex morphological structures, and if such modules enable local response to environmental stimuli.

To begin to address these outstanding questions, we performed spatially resolved RNA-seq on slime mold plasmodia under different growth conditions and single-nucleus RNA-seq on a slime mold plasmodium under different growth conditions at two time points (see *Supplementary file 1* for sample overview). Sequencing reads were mapped against the *Physarum* transcriptome reference (*Glöckner and Marwan, 2017*; *Schaap et al., 2015*) with the data enabling localization of gene expression profiles to different structures within the plasmodium and exploration of local transcriptome responses, which correlated with nuclei heterogeneity. Taken together, our data suggests that nuclei within *Physarum* are mobile and can integrate local signals to coordinate a transcriptional response to dynamic environmental conditions, which enables the syncytium to locally change behavior and morphology.

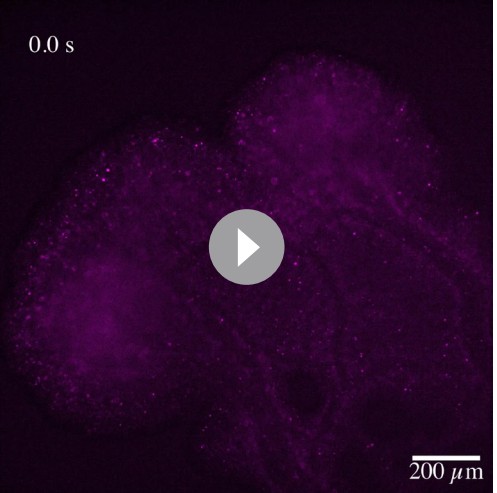

**Video 2.** Nuclei shuttle flow across a fan region. Live recording of SYTO62-stained nuclei in a fan region of a *Physarum* plasmodium over one period of cytoplasmic shuttle streaming. Differences in nuclei sizes originate from the single imaging plane of the three-dimensional tubes of the plasmodium and rarely by contamination of the autofluorescence of larger vesicles.
https://elifesciences.org/articles/69745/figures#video2

## Results
### Nuclei are distributed throughout the slime mold syncytium and can be mobile
We pursued several whole-plasmodium imaging experiments on fixed and live plasmodial cultures to better understand nuclei size, location, and

dynamics within different substructures of the syncytium (*Figure 1B*, *Figure 1—figure supplement 1*). We first grew plasmodial cultures on Phytagel plates, fixed the plasmodium with paraformaldehyde, and stained nuclei with Hoechst 33342. We observed tens of thousands of nuclei distributed across all structures of the plasmodia (*Figure 1C*). We found that the nuclei have an estimated mean size of 5.65 µm and a mean inter-nuclei distance of 6.63 µm (*Figure 1D*). In particular, the maximum distance between two nearest nuclei is 33.9 µm, but 99.9% of the nuclei are within a distance of 12 µm apart from each other. We used a convolutional neural network to segment nuclei (*Weigert et al., 2020*), which revealed a heterogeneous distribution of nuclei across the plasmodium (*Figure 1E*). The mass differences across the plasmodium were used to unbiasedly separate the plasmodium broadly into a growing front region (top) and a primary vein/network part (bottom) by estimating the Otsu threshold (*Figure 1F*; *Otsu, 1979*), and we observed that the growing front region shows significantly increased nuclei densities compared to network veins (*Figure 1G*).

To explore nuclei dynamics in the syncytium, we stained live plasmodia using a fluorescent dye (SYTO62) and imaged vein networks and fan regions in the syncytium over 100 s (*Figure 1H*). Nuclei segmentation (*Weigert et al., 2020*) confirmed a heterogeneous distribution of nuclei, with an increased nuclei density within the growth front of the slime mold similar to what was observed in the fixed plasmodium (*Figure 1I*). Within the same imaging field, we observed immobile and mobile nuclei in close proximity. Mobile nuclei are advected by the local shuttle flow in veins of the network (*Figure 1J*, *Video 1*) and the foraging front (see *Video 2*). Strikingly, advected and immobile nuclei coexist in the syncytium and can get trapped in the ectoplasm – consistent of cross-linked actomyosin fibrils allowing periodic contractions, or be released into the peristaltic flow of the endoplasm that comprises the continuously shuttled part of the cytoplasm (see also *Figure 1B*; *Kamiya, 1950*; *Wolf and Sauer, 1982*). Altogether, these data show that nuclei are distributed throughout the plasmodial syncytium, and that they are maneuvered as structures form and grow.

## Spatial RNA-seq uncovers morphology- and region-specific gene expression profiles in the slime mold syncytium

To determine if there is transcriptional heterogeneity in different domains and structures within the slime mold, we performed spatial RNA-seq measurements across individual slime mold plasmodia (SM1–4, *Figure 2A*, *Figure 2—figure supplement 1A*; see *Figure 2—figure supplement 2* for additional experiments). All slime molds (SM1–4) originated from the same micro-plasmodium culture and were washed with nutrient-free medium to ensure similar starting conditions prior to growth on standard 384-well plate covers coated with a thin agar layer. SM1 and SM2 were collected after 20 hr of growth, whereas for SM3 and SM4, one oat flake was placed 2 cm away from the initiating plasmodium. After an additional 3 hr of culture, growth fronts in both SM3 and SM4 were in close proximity to the oat flake (*Figure 2—figure supplement 1A*). At this point, SM3 was collected for spatial RNA-seq, whereas SM4 was allowed to cover and assimilate the oat and grow for an additional 4 hr prior to collection for spatial RNA-seq (*Figure 2—figure supplement 1A*). Spatially restricted plasmodium sampling was achieved by placing slime molds onto a 384-well plate reservoir followed by centrifugation such that position information was preserved in 4 mm square grids (*Figure 2—figure supplement 1B*). Bulk transcriptome libraries were generated using the SMART-SEQ2 protocol (*Picelli et al., 2013*) with small modifications (see Materials and methods), barcoded, and sequenced, and expression profiles were analyzed and mapped back to grids overlaying the original plasmodial structure (see *Supplementary files 2-6* for details). Spatially registered data from all slime molds were combined and a nonlinear dimensionality reduction by Uniform Manifold Approximation and Projection (UMAP) was performed (*Figure 2B*). Grid clustering revealed distinct expression profiles from each slime mold (*Supplementary file 7*) by identifying cluster-specific genes between slime mold clusters (*Figure 2C*), which did not change when using also unannotated transcripts for the analysis (*Figure 2—figure supplement 1H*). The SM3 grids showed the strongest difference in gene expression compared to the other plasmodia (*Figure 2—figure supplement 1C*). This difference was driven by the expression of genes involved in signal transduction (DHKL) (*Parikh et al., 2010*) and chemotaxis (GEFA) (*Charest et al., 2010*) inferred from studies in the cellular slime mold *Dictyostelium*, and cell shape polarity studied in yeast (TEA1) (*Snaith and Sawin, 2003*; *Figure 2C*). In line with the potential function of these marker genes, Gene Ontology (GO) enrichments for 'regulation of signal transduction' and 'neuropeptide signaling pathway signal transduction' were only observed

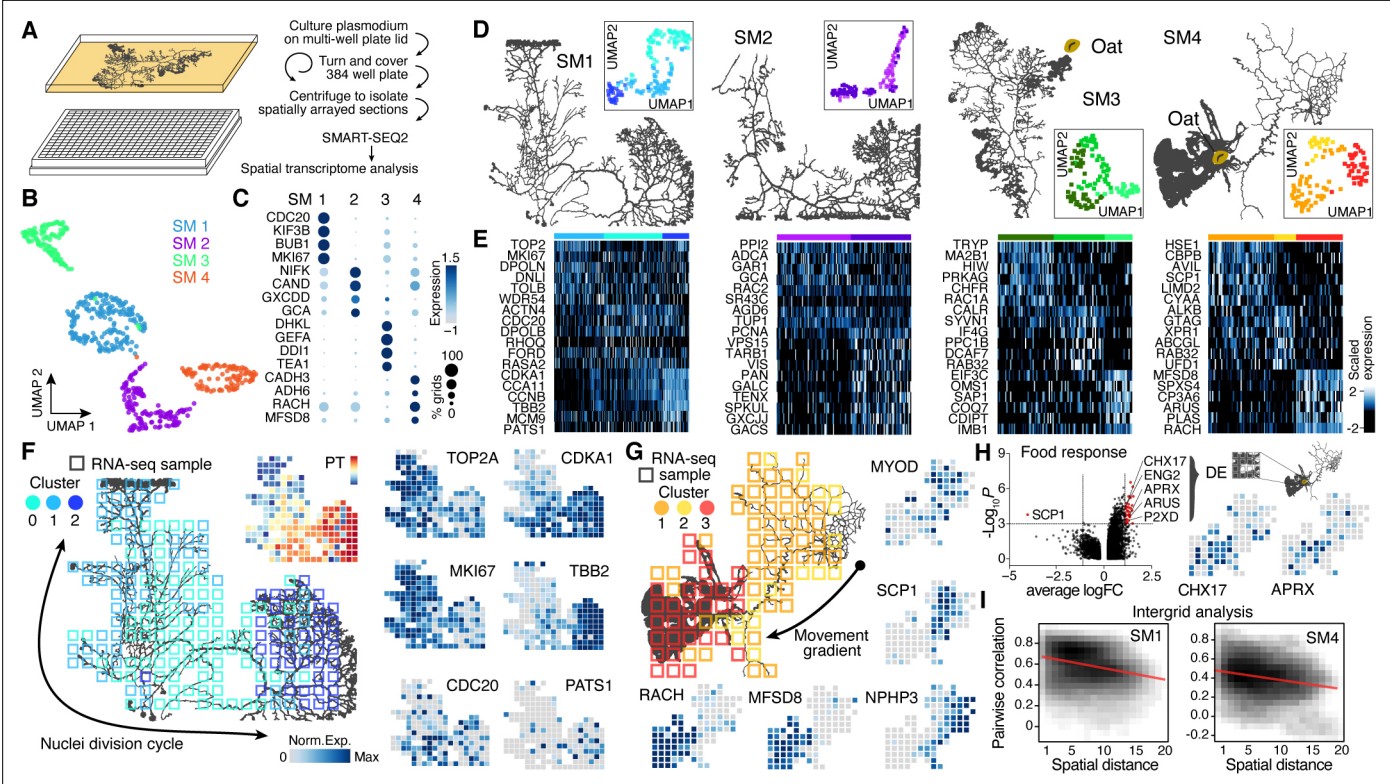

**Figure 2.** Spatial transcriptomics reveals region-specific gene expression in the plasmodial slime mold syncytia. (**A**) Schematic of experimental setup. (**B**) Four slime mold plasmodia were grown overnight from a microplasmodia culture. Uniform Manifold Approximation and Projection (UMAP) embedding reveals general differences between slime molds. (**C**) Dotplot visualizes scaled gene expression intensity (dot color) and frequency (dot size) for top slime mold-specific marker genes, respectively. (**D**) Morphology overview of slime mold plasmodia investigated. SM1 and 2 were directly investigated after grown from microplasmodia culture, whereas oat flakes were supplied to SM3 and 4 and experiments were performed either just before the slime mold reached the food (SM3) or 4 hr after it reached the oat flake and had started re-expanding (SM4). UMAP embeddings next to each plasmodium reveal gene expression differences across grids for individual slime molds. (**E**) Heatmap representations of gene expression intensities for cluster markers (rows) across all grids (columns). Grids are ordered by cluster identity shown in (**D**). (**F**) Sampling grid on the 384-well plate overlaid by the plasmodium of SM1 at the time when sampled (left). Different colors represent the clustering result from (**D**). Right: marker genes from (**E**) are visualized as feature plots onto the sampling grid. (**G**) Sampling grid on the 384-well plate overlaid by the plasmodium of SM4 at the time when sampled (top left). Different colors represent the clustering result from (**D**). Bottom: marker genes from (**E**) are visualized as feature plots onto the sampling grid. (**H**) Volcano plot (left) reveals differentially expressed (DE) genes specific to grids where the plasmodium was in direct contact with a food source (oat flake). Feature plots (right) visualize the spatial arrangement for two of the DE genes of the volcano plot. (**I**) Density plots visualize the relation between the pairwise distance against the pairwise Pearson's correlation across all samples of the grid of SM1 (left) and SM4 (right), respectively. Statistically significant negative correlations are identified and marked by a red line.

The online version of this article includes the following figure supplement(s) for figure 2:

**Figure supplement 1.** Spatial transcriptomics experiments allow to identify gene expression differences across plasmodial slime mold syncytia.

**Figure supplement 2.** Spatial transcriptomics experiments with manual picking of plasmodial pieces.

**Figure supplement 3.** Gene Ontology (GO) analysis of cluster-specific genes across plasmodial slime mold syncytia.

for SM3 (*Figure 2—figure supplement 3* and *Supplementary file 7*). As noted above, a SM3 growth front was actively extending toward the oat flake at the moment of sampling (*Figure 2D*), suggesting that the transcriptomic differences observed in this slime mold are linked to the chemotactic stimulus absent for the other slime molds. Further experiments are needed to understand whether the stimulus is the cause or the consequence of the change in gene expression associated with chemotaxis in SM3. However, shared gene expression profiles for the biological replicates SM1 and SM2 support that similar growth conditions can likely drive similar gene expression programs (*Figure 2—figure supplement 1D*). In addition, batch correction identified genes that were shared across plasmodia, such as RAC1 that is expressed in SM1, SM3, and SM4 or COAA that is expressed in SM1, SM2, and SM4 (*Figure 2—figure supplement 1E–G*). We observed that growth front regions can be identified

by a common set of fan-specific genes for each plasmodia (*Figure 2—figure supplement 1I*) and that fans also cluster together on the transcriptome level with only SM3 being distinct (*Figure 2—figure supplement 1J*). In contrast, all grids of SM1 were enriched in cell cycle-related genes (BUB1, MKI67, KIF3B), and we detect a unique cluster of the GO terms 'Intraciliary transport,' 'Spindle assembly,' 'Cell wall biogenesis,' 'Centromere complex assembly,' and 'Chromosome segregation' only in SM1, suggesting that SM1 was in the process of nuclei division (*Figure 2C*; see also *Figure 2—figure supplement 2D*, *Figure 2—figure supplement 3*, and *Supplementary file 7*). Altogether, these data comparing four slime molds suggest that transcriptome variation can be detected between slime molds grown from the same parent culture, but experiencing different conditions.

We next analyzed whether transcriptome heterogeneity also exists within individual slime mold syncytia. We performed dimensionality reduction for each individual slime mold spatial RNA-seq data and visualized grid heterogeneity using UMAP (*Figure 2D*, *Figure 2—figure supplement 1K and L*). These analyses revealed substantial intra-syncytium heterogeneity, and we identified genes that were specifically expressed in each cluster (*Figure 2E* and *Supplementary file 7*). Strikingly, we observed gene expression patterns that suggested coordinated intra-syncytial behaviors. For example, in SM1 a major source of heterogeneity was determined by differences in mitosis dynamics as certain clusters expressed G2/M phase markers (TOP2A, MKI67), whereas other clusters expressed markers for other phases of the cell cycle. Mapping these transcriptomic profiles onto the respective grids of SM1 revealed a wave of molecular features evidencing nuclei division orchestration across the plasmodium (*Figure 2F*; see also *Figure 2—figure supplement 2E and F*), a phenomenon that was first observed in the beginning of the 20th century but has not been molecularly described (*Howard, 1932*; *Schiebel, 1973*). Portions of the plasmodium show genes associated with G2M transitioning phase (TOP2A, MKI67) (*Tirosh et al., 2016*), whereas the center of the slime mold is characterized by the anaphase promoting gene CDC20 (*Fang et al., 1998*), and the bottom-right portion by spindle and cytokinesis marker genes (CCNB, TBB2, AURAA) (*Roghi et al., 1998*; *Figure 2F*), suggesting that the wave of nuclei divisions initiated in the bottom-right part of the slime mold (cluster 2). Interestingly, the GO terms 'Chromosome segregation' and 'Spindle assembly' follow this wave and are separated by cluster identity in which these terms were enriched (*Figure 2—figure supplement 3B*).

Similarly, when mapping heterogeneity observed in SM4 onto the spatial grids, we found that clusters distinguish morphological structures in the slime mold. For example, grid cluster 3 is associated with a growth front, whereas other clusters overlap the network (*Figure 2G*). We note that certain genes that generally distinguish SM4 from SM1-3, such as alcohol dehydrogenases ADH6 and CADH3 (*Arabidopsis thaliana* orthologs), are frequently and most highly expressed throughout this particular mold and may be involved in metabolizing oat compounds (*Figure 2C*). In agreement with this assumption, the terms 'Glycogen synthetic pathway' and 'Glycolysis' (lumped in *Figure 2—figure supplement 3H*) are only found in SM4 of the four plasmodia tested. Strikingly, there are genes such as a sarcoplasmic calcium-binding protein (SCP1) that are found mostly in invertebrate muscles (*Hermann and Cox, 1995*) or MYOD that is thought to stabilize and even retract cortical structures in *Dictyostelium* (*Jung et al., 1996*) enriched in the SM4 network region, highlighting the role of the network to generate cytoplasmic flows across the plasmodium (*Figure 2G*; *Alim et al., 2013*). The fan region, in contrast, is characterized by the gene racH, which regulates actin filament polymerization in *Dictyostelium* (*Somesh et al., 2006*). The expression of racH is linked with the coexpression of the major facilitator superfamily domain-containing protein 8 (MFSD8), which is a transmembrane carrier that transports solutes by using chemiosmotic ion gradients and plays a role in TORC1 signal transduction that controls the growth rate in yeast (*Kunkel et al., 2019*).

Another interesting aspect of SM4 is that it had recently assimilated an oat flake allowing us to test whether slime mold structures in direct contact with a nutrient source acquire a distinct gene expression profile. We selected a region of 4 × 4 grids covering the oat flake and performed a differential gene expression analysis against all other grids. Intriguingly, this analysis revealed genes with physiological links to nutrient uptake (*Figure 2H*). For example, the putative K+/H+ exchanger CHX17 known to be expressed in *A. thaliana* roots is involved in maintaining pH homeostasis (*Cellier et al., 2004*). Similarly, the extracellular alkaline metalloprotease APRX (*Kodama et al., 2007*) involved in casein degradation by *Pseudomonas* species and other psychrotolerant bacteria (*Maier et al., 2020*) is enriched in the region covering the oat flake. This finding is supported by enrichments for the GO

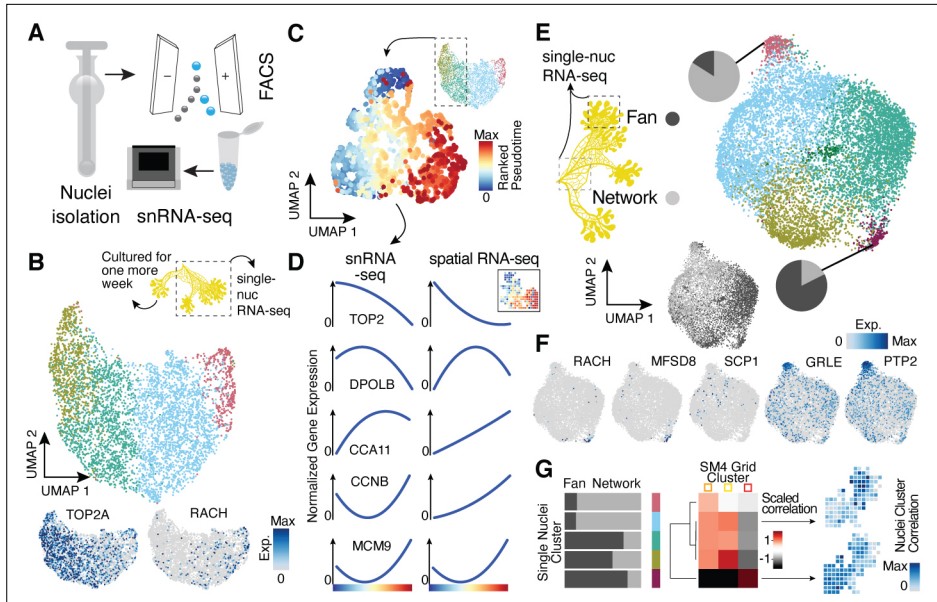

**Figure 3.** Single-nucleus RNA-sequencing uncovers nuclei diversification within the syncytia. (**A**) Nuclei extraction using a douncer with subsequent nuclei enrichment through FACS was performed prior to single-nucleus experiments. (**B**) Uniform Manifold Approximation and Projection (UMAP) embedding visualizes nuclei heterogeneity of a freshly grown plasmodium. Cluster-specific marker genes are visualized as features on the embedding. (**C**) Pseudotemporal ordering of nuclei extracted from clusters with high TOP2A expression in (**B**). Extracted nuclei were re-embedded by UMAP with pseudotime ranks shown on the embedding. (**D**) Gene expression changes of cluster marker (see *Figure 2F*) genes across pseudotime are shown for snRNA-seq (left) and spatially resolved grid RNA-seq data (right). (**E**) UMAP embedding reveals nuclei heterogeneity of a 1-week-old slime mold plasmodium with nuclei originating from different parts of the plasmodium. Pie charts reveal different nuclei proportions of two exemplary clusters. The UMAP inset encodes the origin of each nucleus by color. (**F**) Cluster marker genes for (**E**) are presented as feature plots on the embedding. (**G**) From left to right: bar chart representing proportions of nuclei origin per cluster shown in (**E**). Heatmap representation of scaled correlation values between the pseudobulk gene expression per cluster of the plasmodium in (**E**) and the pseudobulk gene expression per cluster for SM4 (*Figure 2G*). Arrows indicate for which clusters correlations against individual grids of SM4 were estimated with the result being presented as a feature on the grid embedding.

The online version of this article includes the following figure supplement(s) for figure 3:

**Figure supplement 1.** Nuclei heterogeneity revealed by single-nuclei RNA-seq.

terms 'Glycogen synthetic pathway' and 'Glycolysis' in SM4-specific cluster markers (*Figure 2—figure supplement 3H*).

Overall, for SM1, SM2, and SM4, there was a clear trend where the closer grids are in space, the more correlated they are in their transcriptome (*Figure 2I*, *Figure 2—figure supplement 1M and N*). Altogether, this data demonstrates that the *Physarum* syncytium is able to generate localized gene expression patterns that are associated with structures, growth states, and environmental stimuli, and patterns are globally coordinated throughout the noncompartmentalized syncytia.

## Single-nucleus RNA-sequencing uncovers nuclei heterogeneity that correlates with spatial transcriptome dynamics and structure specialization

We next sought to determine if the spatial transcriptome differences observed in the syncytium are apparent at the level of single nuclei. We isolated nuclei from two plasmodia and performed single-nucleus RNA-sequencing (snRNA-seq) using a droplet-based single-cell genomics platform (10x Genomics) (*Figure 3A*). The same plasmodium was sampled twice, first directly after an overnight plasmodium formation from a micro-plasmodium liquid culture (primary) and second after 1 week of culture with oat flakes (secondary, see Materials and methods and *Supplementary file 1*). Additionally, the 1-week secondary plasmodium was separated prior to nuclei isolation into a growth

front-enriched (fan) and a network-enriched (network) sample (*Figure 3—figure supplement 1A*). Syncytial nuclei are small with a diameter of around 5.6 μm (*Figure 1D*), and we therefore detected on average 591 transcripts (UMIs) per nucleus with a homogenous distribution of transcript counts across nuclei subpopulations (*Figure 3—figure supplement 1B*). Unsupervised nuclei transcriptome clustering and visualization of heterogeneity in an UMAP embedding revealed gradients of nuclei diversity rather than clearly distinct states in line with the lack of compartmentalization in the syncytium (*Figure 3B–E*, *Figure 3—figure supplement 1C–E*; *Supplementary file 7*) with unannotated transcripts having only a minor influence on gradient formation (*Figure 3—figure supplement 1F*). In the primary sample, we observed signatures suggesting a wave of nuclei division was in progress at the time of plasmodium sampling (*Figure 3B*) strengthened by the finding that the GO term 'Chromosome segregation' is only found in the cycling plasmodium SM1 and in the primary plasmodium sample. In addition, we noticed a nuclei cluster that specifically expressed racH, a gene we identified to be growth front-specific in the spatially resolved bulk dataset (*Figure 2G*). We extracted two clusters with strong mitosis signatures and ordered these nuclei along a cell cycle pseudotime (*Figure 3C*, *Figure 3—figure supplement 1C*). Comparison of this nuclei pseudotemporal trajectory with a pseudospatial ordering of grids from SM1 revealed strong concordance in wave-like mitotic gene expression signatures (*Figure 3D*). This data supports that spatial gene expression heterogeneity can be observed at the level of single nuclei.

The growth front and network samplings from the secondary plasmodium exhibited slightly more nuclei heterogeneity compared to the primary plasmodium (*Figure 3E*; see also *Figure 3—figure supplement 1D*), which was again unaffected by the mode of input transcripts (all versus annotated, *Figure 3—figure supplement 1G*). We observed unequal mixing of nuclei with certain clusters dominated by nuclei from either the growth front or the network region. Strikingly, the cluster dominated by growth front nuclei is marked by racH, whereas the network-dominated cluster is marked by SCP1 (*Figure 3F*), in line with the spatial transcriptome data. Interestingly, there is nuclei cluster 3 that expresses a glutamate receptor (GRLE) and a tyrosine-protein phosphatase (PTP2) both known to be involved in aggregation during fruiting body (sorocarp) development in the cellular slime mold *Dictyostelium* (*Taniura et al., 2006*; *Ramalingam et al., 1993*; *Figure 3F*). However, we observed neither morphological signs of fruiting body formation nor an enrichment in previously identified sporulation-related genes (*Glöckner and Marwan, 2017*) in this cluster (*Figure 3—figure supplement 1E*). Instead, we find the GO term 'Glycogen synthetic pathway' to be enriched in cluster 3 (*Figure 3—figure supplement 1*) – a term also enriched in the oat-associated plasmodium SM4 (*Figure 2—figure supplement 3H*) and here for the secondary plasmodium in the region in direct contact with oat flakes in our experiment (*Figure 3—figure supplement 1A*). These nuclei may be in a precursor state prior to induction of sporulation or sclerotium formation but clearly show a transcriptomic response to the direct contact to a nutrient recourse.

Motivated by our finding that we observed overlaps of GO terms and exemplary genes between our nuclei data and the spatially-resolved data, we next determined if nuclei populations can be projected to spatial locations. We averaged expression values across single-nuclei clusters and across spatial transcriptome clusters from SM4 (see *Figure 2G*). We calculate correlations between nuclei and spatial clusters, which allowed us to associate nuclei with different regions of the spatially segmented syncytium (*Figure 3G*). Strikingly, the fan-specific cluster 3 in SM4 has the highest correlation with the nuclei cluster 4 that is strongly enriched with nuclei from the fan sample (*Figure 3E*), allowing us to draw a direct link between the nuclei and the spatially resolved data. This result highlights that it is possible to infer the potential location of the nuclei in the plasmodium without knowing its spatial organization based on their transcriptomic state.

Altogether, this data suggests that nuclei within the syncytium are transcriptionally heterogeneous, and that the particular transcriptome state associates with the location within the plasmodium (growth front vs. network). Whether the plasmodium behavior, the environment, intrinsic cues, or a mixture of these components lead to the diversification of nuclei states and vice versa needs to be further explored.

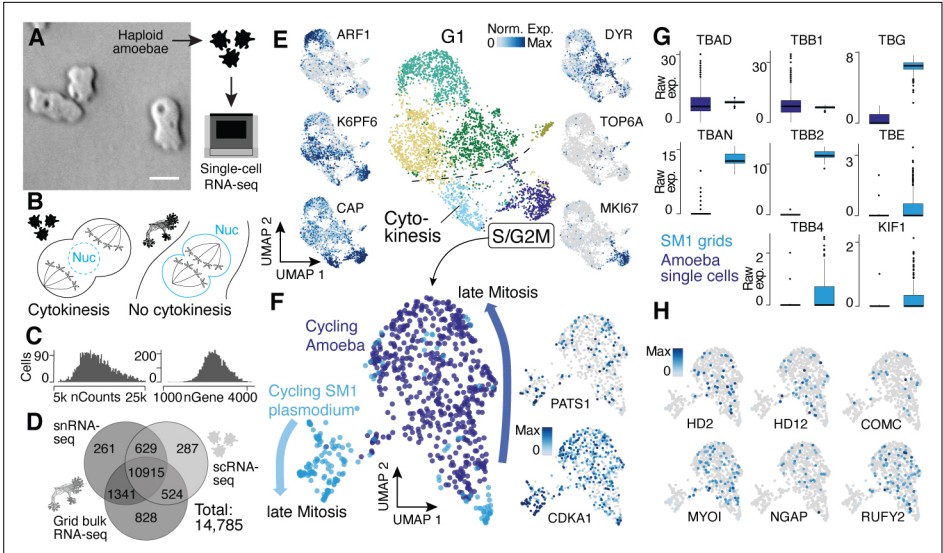

**Figure 4.** Single-nucleus amoebae are heterogeneous and differentially regulate cytokinetic programs compared with syncytial nuclei. (**A**) Brightfield image of amoebae (single frame from *Video 3*). Scale bar: 10 μm. (**B**) Schematic illustrating mechanistic differences during mitosis in amoebae (left) and plasmodia (right). (**C**) Histograms reveal the distribution of the raw transcript counts per gene (left) and the number of different genes per cell detected (right) for amoeba scRNA-seq data. (**D**) Venn diagram shows the number of genes detected per experimental setup and their overlap. A 10% quantile cutoff was applied to remove sparsely expressed genes for the comparison. (**E**) Uniform Manifold Approximation and Projection (UMAP) embedding of single haploid amoeba cells (center). Cell cycle stages are indicated and gene expression intensities are visualized as features on the embedding (left/right). (**F**) UMAP embedding of single amoeba cells in G2M phase and samples of a cycling plasmodium (SM1). RNA-seq samples were integrated using Seurat's built-in CCA 'anchoring' (*Stuart et al., 2019*) to allow a comparison between single-cell (10x Genomics) and bulk RNA-seq (SmartSeq2) data. Feature plots of PATS1 and CDKA1 reveal cell/nuclei cycle directionality for samples. Drawn arrows reveal directionality. (**G**) Cell cycle-related marker gene expression identified through cluster-specific gene expression analysis between amoeba cells and plasmodium grids in (**F**) is visualized as a boxplot for plasmodium and amoeba samples, respectively. TBAD and TBB1 are shown as reference for non-DE genes. (**H**) Amoeba-specific genes are visualized as feature plots on the UMAP embedding in (**F**).

The online version of this article includes the following figure supplement(s) for figure 4:

**Figure supplement 1.** Analysis of the impact of unannotated transcripts on amoeba cell heterogeneity.

## Single-nucleus amoeba are heterogeneous and express genes distinct from nuclei within the single-celled syncytium

We next wanted to understand if the different nuclei states detected in the plasmodial syncytia are also observed in the single-nucleus *Physarum* amoeba stage and whether there are transcriptomic differences originating from the different nucleic division cycles (*Figure 4A and B*). Toward this goal, we characterized haploid *Physarum* amoeba cells using single-cell RNA-seq (*Figure 4A*). We detected on average 14,525 mRNA molecules per cell (*Figure 4C*), which is a higher value than for the snRNA-seq data because of the mRNA-rich cytoplasm not sampled in snRNA-seq experiments. However, we neither detect a greater diversity nor more unique genes for the amoeba compared to the snRNA-seq plasmodium data (*Figure 4D*); instead, we see that the diversity is increased for the grid bulk RNA-seq samples with a larger overlap with the plasmodium snRNA-seq data than the amoeba data. The same proportions are found when using all transcripts of the transcriptome for the same comparison (*Figure 4—figure supplement 1A*). This shows that amoeba do not use a large set of unique genes, whereas the plasmodium uses an increased repertoire of genes from its genome. We next explored the diversity of amoeba states and observed substantial variation in amoeba transcriptome states, identifying at least six distinct clusters at this resolution, with most of the heterogeneity being linked to cell cycle kinetics and subsequent G1 phases (*Noegel et al., 1999*; *Figure 4E*; *Supplementary file 7*), which is different to the plasmodium data with all nuclei being in roughly the same G1 phase due

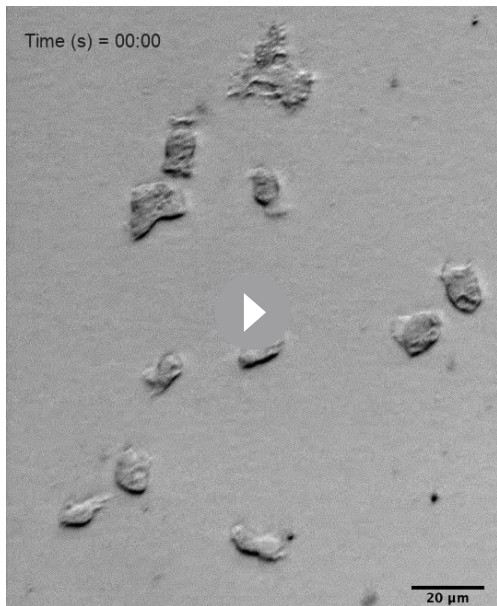

Time (s) = 00:00

20 µm

**Video 3.** Living amoeba.
 Brightfield live recording of haploid single-nucleus *Physarum* amoebae.
https://elifesciences.org/articles/69745/figures#video3

to the synchronous nuclei division. Using all genes and not only annotated transcripts did not have a major impact on the assessment of amoeba cellular diversity (*Figure 4—figure supplement 1B*). Interestingly, one cell cluster was marked by a topoisomerase family six member (TOP6A) (see *Figure 4—figure supplement 1C* for other markers), a gene family playing a major role in eukaryotic meiosis (*Ramesh et al., 2005*; *Dee, 1966*). These data illuminate amoeba cell states and reveal that substantial variation in cell states coexists within a *Physarum* single-cell culture.

An interesting feature of the multinucleate syncytia is that the mitotic spindle forms within the nucleus to segregate chromosomes into two daughter nuclei, which is common among syncytial fungi ('closed mitosis,' *Figure 4B*; *Solnica-Krezel et al., 1991*; *Tanaka, 1973*; *Heath, 1980*). In contrast, *P. polycephalum* amoebae anchor the mitotic spindle through centrosomes at the opposing poles of cells ('open/astral mitosis') as occurs during typical eukaryotic cell divisions (*Figure 4B*; *Solnica-Krezel et al., 1991*; *Havercroft and Gull, 1983*). We therefore integrated spatial transcriptomes from the cycling syncytium (SM1) and transcriptomes from amoeba mitotic cells and explored the expression similarities and differences. We observed genes that increase expression toward late mitotic phases in both *Physarum* states (*Figure 4F*), for example, CDKA1 and PATS1, which are involved in cytoskeleton dynamics during cytokinesis (*Sofroni et al., 2020*; *Abysalh et al., 2003*). However, we observed striking differences in other gene expression profiles. For example, certain tubulins and one kinesin – major components of the spindle assembly pathway – are differentially expressed between the plasmodium and amoebae (*Figure 4G*). We find that alpha tubulin 1a (TBAD) and beta tubulin 1 (TBB1) are spindle markers expressed in both states, whereas alpha tubulin 1b (TBAN) and beta tubulin 2 and 4 (TBB2, TBB4) are only detected at high levels in the plasmodium, consistent with previous reports (*Burland et al., 1988*; *Burland et al., 1983*). Also, the epsilon and gamma tubulin chains (TBE, TBG) involved in centrosome duplication and spindle formation and the spindle-associated kinesin KIF1 are detected at much higher levels in the plasmodium. Other spindle components such as dyneins do not show a state-specific expression. These data suggest that the differential usage of tubulins orchestrates a change from an open mitosis with cytokinesis in amoeba, to a closed mitosis with nuclei division in the plasmodia, resulting in a syncytia.

In line with this finding, we note that all cell cycle-related genes detected in the amoeba are also expressed in the plasmodial stage, suggesting that the open mitosis in *Physarum* amoeba is accomplished by the same proteins also active in the plasmodium state but that the mechanisms could be different. Instead, we observe a specific expression of genes related to the amoeba lifestyle (*Figure 4H*) with some genes being well studied in the amoeboid cellular slime mold *Dictyostelium* such as MYLI involved in phagocytosis and filopodia assembly (*Tuxworth et al., 2001*; *Titus, 1999*), COMC, a gene important for intercellular communication (*Kibler et al., 2003*), or the RAS GTPase activating gene NGAP, which is involved in chemotaxis in *Dictyostelium* (*Xu et al., 2017*). In addition, multiple transcription factor-like genes (HD2, HD12, RUFY2) were identified. Together, these data reveal similarities and differences in transcriptome state regulation in amoebal and plasmodial phases of the *Physarum* life cycle. Nevertheless, additional validations and experiments are needed to fully understand the interplay of differential usage of genes and the potential rearrangements of shared

proteins between amoeba and plasmodia that leads to a switch in mitotic states when a plasmodium forms.

## Discussion

Multicellular organisms achieve an amazing complexity of forms and functions through the compartmentalization and specialization of different parts of the organism. However, the slime mold *P. polycephalum* is an extraordinary example of an ostensibly unicellular organism that exhibits specialized morphologies and behaviors across its life cycle. The *Physarum* genome encodes instructions to generate and maintain both single-nucleus amoeboid cells as well as the gigantic single syncytial cell that develops complex structures and exhibits intricate behaviors. Indeed, *P. polycephalum* can be translated as the 'many-headed' slime, which describes the remarkable ability of *Physarum* to develop multiple foraging growth fronts to sample the environment and seek out new terrain. This structural and behavioral complexity raises the questions around how such specialization and coordination can be generated, maintained, and quickly adapted to changing environmental conditions?

Our results suggest that *Physarum*'s dynamic structure formation and response to local environmental conditions are coordinated in part through localized gene regulation. We observed striking spatial differences in gene expression profiles within individual syncytial cells. These patterns appeared to distinguish different structural components of the plasmodium and also correlated with environmental stimuli (e.g., nutrient source) at individual growth fronts. Our view is that genomic DNA partitioned into nuclei are distributed throughout the syncytium and serve as local processing units that receive local and systemic input signals to coordinate encoded outputs. Nuclei can be locally fixed in the ectoplasm and remain in the same position for extended time periods of at least several minutes (extrapolation inferred from *Videos 1 and 2*), or they can be advected over short or long distances through the endoplasmic shuttle flow as the plasmodium pulsates and grows. This system enables the syncytium to establish complex behavioral responses that can vary in distant parts of the slime mold.

Intriguingly, we find that there is transcriptome heterogeneity among nuclei within the same plasmodium. Single-nuclei analysis revealed different proportions of nuclear transcriptome states in the growth front and fan regions. We hypothesize that it could be beneficial to maintain different nuclei states for rapid access to different transcriptional programs at any time. These data suggest that one mechanism for rapid and diverse response might occur through the co-maintenance of multiple specialized or randomized epigenetic states. We do not have evidence that the syncytium is an assembly of intrinsically specialized nuclei that exist prior to plasmodium formations as we do not find clear evidence of linked specialized states within *Physarum* amoebae. However, a future area of research will be to understand if transcriptome states are propagated through nuclei divisions such that outputs can be maintained rapidly in the local growth front or network microenvironment. In addition, consistent with previous reports (*Earnshaw and Steer, 1983*), we observed many intact nuclei within the endoplasm. These nuclei shuttle at rapid speeds of up to 1.3 m/s (*Kamiya, 1950*) throughout the plasmodium within a few contraction periods (*Alim et al., 2013*; *Nakagaki et al., 2000b*). Yet, it is unclear whether such a shuttle system could be used to 'seed' nuclei states into particular locations within the mold. Interestingly, the effective dispersion coefficient is four times smaller in the fan region as compared to the core network (*Marbach et al., 2016*) with the highly branched fan region hindering effective long-distance transports (comparing *Videos 1 and 2*). As a consequence, once nuclei are shuttled to a fan region their local dwelling time is relatively extended and the nuclei are thus more likely to get trapped within the foraging front. This suggests that nuclei from other plasmodial locations (e.g., close to a specific food resource) when transported to the new foraging front have the potential to 'seed' more quickly an adequate gene expression module when needed in the newly explored environment. That stated, our data strongly support nuclear adaptation to local and temporally variable environmental stimuli but further tests are needed to prove our hypothesis that shuttled nuclei can 'seed' nuclei states at new locations.

Our comparative analysis of single-nucleus amoeba and syncytial cell states reveals that mechanisms for different types of nuclei division are encoded in the *Physarum* genome. Interestingly, plasmodium-specific gene sets are additionally used to accomplish the nuclei division in the syncytium, whereas a shared core set of mitotic genes is expressed in amoebae and the plasmodium. This suggests that the switch in mitotic behavior is partly achieved through an activation of an additional genetic program in the plasmodium that most likely results in a mechanistic change of how the core

set of mitotic gene products is organized. Understanding how this switch is triggered and how the reorganization is mechanistically achieved must be the basis of future research.

The resulting wave of synchronized nuclei divisions in the plasmodium is of particular interest as it is orchestrated across a gigantic syncytium, which makes it a unique model organism for studying syncytial behavior including the mode of endoreplication compared to other syncytial types as observed in fungi hyphae or fly development. Even within plasmodial slime molds, different modes of endoreplication, that is, synchronous or asynchronous (*Kerr, 1988*), exist, but the underlying process is still poorly understood. Notably mitotic synchronicity breaks down once networks exceed a critical size (*Guttes et al., 1961*) or after approximately five generations of mitosis (*Mohberg, 1974*), or when cytoplasmic flows are interrupted (*Wolf et al., 1979*; *Wolf and Sauer, 1982*), suggesting a tight, yet unknown connection between network structure and mitosis.

Generally, the syncytial strategy as is observed in the plasmodial state may be advantageous in certain scenarios, also for cell types within multicellular organisms including humans (*Gandarillas et al., 2018*). Endoreplication supports rapid cellular and organismal growth and studies in plants suggest that multinucleation and the syncytial strategy may enhance plasticity in response to environmental stress (*Cookson et al., 2006*). Our data adds to this view by revealing that individual nuclei can process local environmental signals under variable environmental conditions while being shuttled, and that maintaining a pool of nuclei states at all locations could provide rapid and stochastic access to genetic programs at any location within the syncytium.

Our work is not without certain limitations. Though powerful, the *Physarum* genome and transcriptome have not yet been curated and annotated to the extent of some other highly used model systems. In our analyses, we opted to focus only on the transcripts that have been annotated through comparisons with likely orthologs in other species. Therefore, there are many hundreds of transcripts/proteins that we did not consider here in our analyses of spatial and nuclei heterogeneity. There could be new and exciting biology in the proteins and other functional elements of the DNA that lack homology-based annotation in the *Physarum* genome and transcriptome. In some cases, we rely on very distantly related organisms, such as yeasts, to interpret the markers of heterogeneity in selected ortholog comparisons or in gene ontology enrichments. For example, translating yeast GO annotations (e.g., GO categories chromosome segregation or spindle assembly) to *Physarum* is based on the assumption that many of these genes are involved in the same processes in both species. Similarly, we suggest that there could be meiotic amoebae based on the expression of TOP6A, which is a meiosis marker in other eukaryotic organisms. However, many of these observations are currently speculative and require further research. Genomic and functional comparisons between closely and distantly related species can in the future help to clarify the relationships that we interpret in this study.

Our spatial transcriptomes were generated from four slime molds growing under variable conditions. Unharmonized analysis of the spatial transcriptomic data showed separation of the individual slime molds, which we relate to particular growth features within each individual. However, we highlight that some of this separation is likely due to technical variation. Integration using Harmony and Seurat anchoring indeed harmonizes the grids derived from the different individuals, which also uncovered interesting spatial heterogeneity shared across the specimens. In the future, our spatial profiling methods could be used to explore many *Physarum* individuals grown under different conditions to further assess the influence of the environment on plasmodia. Finally, we associate transcriptome states to plasmodial structure or behavior, which are correlative and do not prove causation. Genetic tools to manipulate plasmodial genes or experiments to physically translocate regions or otherwise modify inputs within the same syncytium could help with interpreting transcriptional changes. Higher resolution in situ spatial transcriptome techniques might be applied to *Physarum* to better localize nuclei states and also determine if there are quiescent and transcriptionally inactive nuclei in the plasmodium.

Altogether, our work reveals that slime molds such as *Physarum* offer the extraordinary opportunity to explore how organisms can evolve regulatory mechanisms to divide labor, switch between different modes of mitosis, specialize, balance competition with cooperation, and perform other foundational principles that govern the logic of life.

## Materials and methods

### Growth and culture of *P. polycephalum*

*P. polycephalum* was cultured as microplasmodia in 100 ml liquid culture of 50% semi-defined medium (SDM) and 50% balanced salt solution (BSS), together with hemin, streptomycin, and penicillin, while shaking at 180 rpm in a 25°C incubator (*Bäuerle et al., 2017*). Change of medium was carried out three times a week.

Axenic *Physarum* amoeba culture was grown in 50 ml liquid shaking culture of SDM at 150 rpm and 30°C. Subculturing happens three times a week to reach a seeding density of $0.2 \times 10^6$ cells/ml (*Nickoloff, 1995*).

### Sample preparation for fluorescence imaging

For fluorescence imaging, a less autofluorescent substrate, 1.2% w/v of non-nutritious Phytagel, was used and distributed in Petri dishes (Ø = 100 mm). Small amounts of microplasmodia (1 ml) were inoculated onto Petri dishes and incubated over 16–24 hr in order to grow small 0.5–1.0-cm-sized plasmodial networks.

For live imaging, nuclei staining was performed by immersing the specimen with 5 µM SYTO 62 (Thermo Fisher S11344) dye for 30 min and washing with BSS solution. The excitation and emission spectrum (649/680) of this dye is far from the autofluorescence spectrum (ex/em 300-450/400-550).

To test the performance of SYTO dyes, a colocalization experiment on fixed samples was carried out (*Figure 1—figure supplement 1A and B*). Slime molds were first grown with the described procedure. For fixation, 2 ml of 4% PFA solution was added on top of the plasmodium with an incubation time of 15 min. The molds were washed two times with autoclaved BSS solution and immersed in a solution of 2.5 µM Hoechst 33342 (Thermo Fisher 62249) and 0.5 µM SYTOX Deep Red (Thermo Fisher S11380) for 30 min. The ex/em ranges of the dyes are 361/497 and 660/682, respectively.

To map the nuclei density of whole plasmodia, *Physarum* was grown from microplasmodia between a thin Phytagel sheet and the Petri dish bottom. This is to limit and homogenize the height of the plasmodia for better comparison of nuclei densities. A blade was used to run along the perimeter of the Phytagel overlay to release the thin sheet from the bottom, before placing the microplasmodia under it. To ensure the immersion of fixation solution and dye will reach the specimen, the staining protocol was also adjusted. PFA solution (4%, 2 ml) was placed on top of the Phytagel, and the Petri dish was shaked on a tilting bed (Polymax 1040, Heidolph Instruments) at 2 rpm for 45 min, before washing it three times with BSS solution. For fluorescent labeling, approximately 200 µl of 2.5 µM Hoechst 33342 was used. The stained specimen was first placed on the tilting bed for 30 min at room temperature and subsequently stored in the incubator at 25°C overnight. Finally, the slime mold was washed twice with BSS.

### DAPI sample preparation

Slime molds were grown on regular agar plates as described above. 4% of PFA solution was added on top of the plasmodium with an incubation time of 30 min. The molds were washed two times with autoclaved distilled water, and DAPI (Thermo Fisher D1306) (300 nM) was added on top. The molds were incubated for 10 min with the staining solution. The solution was then removed, and the molds were washed 2–3 times with autoclaved distilled water.

### Fluorescence imaging

Microscopy images were acquired with the Zeiss Axio Zoom V16 stereo microscope with a Zeiss Plan Neofluar Z 2.3×/0.57 objective, an HXP 120C fluorescence lamp and a Hamamatsu Orca Flash 4.0 V2 complementary metal-oxide semiconductor (CMOS) camera. We used ZEN Blue Edition (Zeiss) to control the microscope. To image Hoechst and red SYTO dyes, filter cubes 02 (BP 365/FT 395/LP 420) and 50 (BP 640/FT 660/BP 690) were used, respectively. For reference autofluorescence imaging, filter cube 38HE (BP 470/FT 495/BF 525) was used. To improve z-resolution, we use the Zeiss ApoTome.

3D micrographs of whole living and fixed cells were acquired directly after staining by scanning over the organism using a motorized xy stage. For every tile of the micrograph, a bottom-to-top z scan was performed over the fluorescent and autofluorescent channels, with a z-step of 2–5 µm and a magnification of 65–258×.

## DAPI imaging

Images were acquired on a Leica TCS SP5 with the following settings: laser power at 29.08, 10.0 × 0.30 DRY HC PL FLUOTAR objective with a numerical aperture (NA) of 0.3 and UV lens FW 20×/0.70.

Images were generated directly after DAPI staining and were acquired from the bottom through microscopy slide and agar layer. DAPI was excited at 405 nm and measured with the filter set Leica/DAPI 430/550. The scan speed was at 200 ms, the pinhole at $7.07 \times 10^{-5,}$ and the pinhole airy at 0.99. The zoom-in was performed by increasing the dbl zoom property from 1 to 3 at plasmodial regions of interest.

## Image processing

A custom written Fiji macro was used to deconvolve the z-stack of the SYTO channel for each tile of the micrograph using the Fiji plugin deconvolutionlab2 (*Sage et al., 2017*) and a generated point-spread-function using PSF-Generator (*Kirshner et al., 2013*). After deconvolving the z-stacks, we maximum-intensity projected the images and used a convolutional neural network-based StarDist-model (*Weigert et al., 2020*) to predict the position and shape of nuclei within the micrographs. Accuracy of the model was tested using a ground-truth dataset training of a Weka-classifier (*Arganda-Carreras et al., 2017*) with subsequent curation of the segmentation. We find an overall precision (TP/(TP + FP)) of 94% and accuracy of 81% (TP/(TP + FP + FN)), indicating a slight underestimation of particle number.

Using the Microscopy Image Stitching Tool (MIST) (*Chalfoun et al., 2017*), we stitched the tiles of the autofluorescence micrograph together, which serves as a reference stitching scheme for the fluorescence micrographs and the segmented images.

The resulting label image of the fluorescent spots was analyszd for the object area, eccentricity, and diameter. Furthermore, we used the detected spot positions to analyze internuclear distance using a Ball-Tree algorithm and perform a kernel density estimation (KDE). The resulting probability density function was integrated over small regions to measure the nuclei density within the slime mold (*Figure 1—figure supplement 1C*).

## Growth and preparation of the plasmodium for sequencing

The slime molds used for sequencing were all provided by the lab of Karen Alim at the Max Planck Institute for Dynamics and Self-Organization (Göttingen, Germany). All slime mold plasmodia originated from microplasmodia liquid cultures as described recently (*Bäuerle et al., 2017*). Briefly, plates had a small cavity in the agar where a small volume of microplasmodia culture was dispensed. In most of the cases, the microsplasmodia needed around 1 day to generate a network structure and expanding as plasmodia outside of the cavity. Oat flakes were added to the plates 1 or 2 days after if the experimental design required it.

## Preparation of nuclei and cell suspensions for 10x Genomics experiments

Samples were kept on ice during all steps of the nuclei preparation. Slime molds were incubated in 2 ml homogenization buffer (10 mM $CaCl_2$, 0.1% Nonidet P-40, 10 mM Tris-HCl, pH 8, 0.25 M sucrose) for 5 min. In addition, the suspension was pipetted up and down a few times to help dissociating the tissue. Afterward, samples were transferred to a douncer and 15 strokes with pestle A and another 15 strokes using pestle B were applied. A 100 µm strainer was used to filter the suspension and was afterward flushed with 2 ml HBSS to collect the remaining nuclei. Nuclei were spun at 300 × g for 5 min. After removing the supernatant, nuclei were resuspended in 500 µl HBSS and filtered through a 20 µm strainer. Another 500 µl HBSS was used to wash off the remaining nuclei from the filter membrane. DAPI (1:1000) was added to the nuclei suspension, and FACS (85 µm nozzle) was performed to isolate single nuclei. 85,000 single nuclei were collected in a well of a 96-cell-culture plate and the concentration of the nuclei, ranging between 200 and 300 nuclei/µl, was estimated using a hemocytometer. FACS of nuclei extracted from amoeba was not possible due to the sensitivity of the nuclei, and the single-cell suspension of amoeba was therefore directly used for 10x Genomics experiments. Cells or nuclei were loaded onto the 10× cartridge (Chip B) at various concentrations, with loading the maximum volume possible for nuclei suspensions or targeting 6000 cells for the

amoeba suspensions. Cell encapsulation, cDNA generation, and preamplification as well as library preparation were performed by using the Chromium Single Cell 3′ v3 reagent kit according to the kit protocol.

## Spatial transcriptomics of plasmodia

100 µl of microplasmodia were plated on a lid of a Corning 384-well Clear Bottom Polystyrene Microplates after spreading a thin layer of agar across the inner side of the lid. The inner side of the lid was covered with parafilm and incubated at room temperature in the dark for 20 hr. On the next day, the 384-well plate was prepared by adding 20 µl lysis solution (6 M guanidine hydrochloride, 1% Triton X-100) in each well (adapted from *Wollny et al., 2016*). Lids with plasmodia spanning a large area of the lid were placed on the 384-well plate, and samples were collected by spinning the plate at 4000 × *g* for 2 min. After sample collection, the plate was placed on dry ice until further processed or frozen at –80°. Note that the collection of slime mold pieces for the initial test experiments (*Figure 2—figure supplement 2*) happened manually by sampling individual 2 × 2 mm grids of three different plasmodia one after another (*Figure 2—figure supplement 2B*). A 96-well PCR plate (Eppendorf) was prepared by adding 13 µl of Agencourt RNA XP SPRI beads to as many wells as needed. RNA was extracted from the lysis mix by transferring 5 µl of lysis mix from the 384-well plate to the 96-well plate prepared with RNA XP SPRI beads (bead to sample ratio of 2.6×; vol:vol) (see *Supplementary files 2-6*). The suspension was incubated for 5 min at room temperature. After binding beads at the tube walls using a magnetic rack, the supernatant was removed. After two rounds of washing with 80% ethanol, RNA was released from the beads by resuspending them in 10 µl EB buffer. Drying the beads was not needed, and the purified RNA suspension was transferred to a fresh 96-well plate and stored at –80° until further processed. 2 µl of this bulk RNA was used as input for SmartSeq2 (*Picelli et al., 2013*). Generation of cDNA followed SmartSeq2 instructions with 16× PCR cycles for cDNA amplification. A high-throughput electrophoresis-based fragment analyzer (Fragment Analyzer, Advanced Analytical Technologies) was used to assess the cDNA fragment size distribution and its concentration of exemplary samples. Illumina libraries were constructed by using the Illumina Nextera XT DNA sample preparation kit according to the protocol. Up to 192 libraries were pooled (3 µl each) and purified with Agencourt SPRI select beads (see *Supplementary files 2-6*).

## Sequencing

Library concentration and size distribution were assessed on an Agilent Bioanalyzer and with Qubit double-stranded DNA high-sensitivity assay kits and a Qubit 2.0 fluorometer for all sequencing libraries. 10x Genomics sequencing libraries were sequenced following the 10x Genomics protocol to a depth of 50,000–100,000 reads per nucleus or cell. Base calling and demultiplexing of single nuclei or cells were performed by using 10x Genomics Cell Ranger 3.1 software. SmartSeq2 libraries were paired-end sequenced (100 base reads) on an Illumina HiSeq 2500 aiming for a depth of ~200,000 reads per sample, base calling was performed using Bustard (*Kao et al., 2009*), and adaptor trimming and demultiplexing were performed as described previously (*Renaud et al., 2015*).

## Data overview

### Spatial transcriptomics

In total, seven slime mold plasmodia were analyzed acting as biological replicates. In addition, each plasmodium was sampled multiple times (100–200 per plasmodium, total 739 independent measures) with each grid representing individual biological replicates of each plasmodium, respectively.

### snRNA-seq data

The same slime mold plasmodium was sampled twice on different days (primary/secondary). Each nucleus per plasmodium (primary: 5292/ secondary: 10330) represents an independent measure providing many biological replicates per plasmodium, respectively. In addition, the transcriptomic data for the primary plasmodium comprises two technical replicates as the same nuclei suspension was processed in two reactions in parallel throughout data generation.

## scRNA-seq amoeba

One amoeba culture was processed with two technical replicates where the same cell suspension was run in two reactions in parallel and with each cell representing a biological replicate of the amoeba culture.

## Data analysis

All sequenced datasets were aligned to the latest version of *Physarum*'s transcriptome (*Glöckner and Marwan, 2017*). All raw reads were aligned using STAR (*Dobin et al., 2013*). Mapping of reads of the 10x Genomics datasets was performed using Cell Ranger 3.1 implemented in the 10x Genomics analysis software, generating absolute transcript counts based on unique molecular identifiers (UMIs). Reads obtained for the SmartSeq2 datasets were mapped with a local installation of STAR, and TPM values were estimated using custom Perl and Bash scripts. Customized RStudio scripts (https://rstudio.com) were used for final preprocessing steps and to run all data analyses. The R packages 'Seurat' v3 (*Stuart et al., 2019*) and 'ggplot' were used for almost all analyses and visualizations unless stated differently. For all feature plots created by Seurat, feature-specific contrast levels based on quantiles of nonzero expression were calculated (q10/q90 cutoff).

## Spatially resolved bulk RNA-seq analysis

SmartSeq2 RNA-seq grid datasets were first merged per plasmodium, and $\log_2$ TPM values were calculated by transforming all values with TPM > 1 and setting all values with TPM < 1–0. Grids with small read coverage (paired reads SM1 < 200k, SM2−4 < 100k) or no clear overlap with the underlying plasmodium network were removed. Only transcripts with a UniProt annotation or at least a gene description in the annotation file (Supplementary file 1 in *Glöckner and Marwan, 2017*) were kept for further analyses while removing all other genes from the grid/gene matrices. This filtering left us with 15,120 of the predicted 28,139 transcripts/genes in the transcriptome that was used for mapping (*Glöckner and Marwan, 2017*). When unannotated transcripts are not removed 27,781 transcripts remained for the analysis. The following analysis descriptions focus on the 'only annotated transcript' objects if not stated differently. Each dataset was then loaded and processed separately using Seurat (Seurat's default normalization was omitted), and data scaling was performed with nGene and nRead regression. Dimensionality reduction consisted of a principal components analysis (PCA) and UMAP with the first 5–15 (SM1:15, SM2:5, SM3:5, SM4:15, T1:10, T2:10, T3:5) principal components (PCs). Louvain clustering (Seurat default) was performed for the same number of input PCs with a resolution of 0.1–0.6 (SM1:0.4, SM2:0.1, SM3:0.3, SM4:0.2, allT:0.6). For plasmodium SM3, SM4 and T1, Louvain clustering gave no convincing results, and we therefore hierarchical clustered grids using hclust's Ward method with distances estimated by Pearson's correlation and the same input gene sets as described above. Differential gene expression (DE) analysis was performed using the Wilcoxon Rank Sum test (Seurat default) on identified clusters or on a selected region for SM4 (*Figure 2H*). The DE genes identified for the oat flake-covered grids were visualized as volcano plot using the R package 'EnhancedVolcano' (https://github.com/kevinblighe/EnhancedVolcano; *Blighe et al., 2021*). In addition, fan scores were calculated for SM1 to SM4 (*Figure 2—figure supplement 1E*), respectively, by scoring each individual grid against the top six marker genes for cluster 3 in SM4 (see *Figure 2E*). Top 10 scoring grids per slime mold were extracted, and a PCA was performed on those 40 grids and afterward embedded in a UMAP by using the first five PCs as input. Grids were hierarchical clustered as described above (*Figure 2—figure supplement 1F*).

For the density plots of grid distance against transcriptome correlation, we selected the top 200 variable genes per individual plasmodium and extracted the scaled matrix (genes/grids) using these genes from each plasmodium. The scaled values were converted into purely positive values by adding the absolute minimum of a gene to the scaled values across all grids for a given gene, respectively. The obtained matrix was then correlated against itself by ignoring diagonal grid correlations. Distances between grids were calculated within the Cartesian scatter by using the R package 'raster.' The density plots were generated using R's smoothScatter() function by plotting grid's densities against transcriptome correlations. The regression line was calculated as a linear model.

Seurat objects were afterward merged to obtain a combined dataset, and dimensionality reduction was performed as described above with five input PCs for the UMAP embedding and Louvain clustering with a resolution of 0.2 (*Figure 2B*). Differential gene expression analysis was performed using

the Wilcoxon Rank Sum test (Seurat default) on all versus all clusters identified roughly representing the different slime molds (*Supplementary file 7*). The same procedure was applied to the Seurat object containing all transcripts, and cluster information was transferred from the 'only annotated' object to allow a comparison of marker genes identified (*Supplementary file 7* and *Figure 2—figure supplement 1H*). Cluster-specific markers are visualized as dotplot (*Figure 2C*) by running Seurat's dotplot function. Another set of differentially expressed genes was obtained by comparing the biological replicates SM1 and SM2 against the two plasmodia (SM3–4) where an oat flake was added to the culture (see *Supplementary file 7* and *Figure 2—figure supplement 1D*). In addition, we used Harmony (*Korsunsky et al., 2019*) and CCA anchoring (*Stuart et al., 2019*) to integrate the data and remove potential batch effects. The first 20 PCs were used as input for both methods and also to generate the UMAP and clusters with standard Seurat functions after data integration (*Figure 2—figure supplement 1E*). The methods resulted in different clusters (*Figure 2—figure supplement 1F*) with Harmony outputting better pan slime mold cluster markers as revealed as features on the UMAP embedding (*Figure 2—figure supplement 1G*).

## Analysis of snRNA-seq plasmodium data

Only transcripts with a UniProt annotation or at least a gene description in the UniProt annotation file by *Glöckner and Marwan, 2017* were kept for further analyses while removing all other genes from the nuclei/gene matrix. 14,055 of the predicted 28,139 transcripts/genes in the transcriptome remained for the primary plasmodium and 14,483 for the secondary plasmodium. When unannotated transcripts are not removed, 24,592 transcripts remained for the primary and 25,815 for the secondary plasmodium. The following analysis descriptions focus on the 'only annotated transcript' objects if not stated differently. Afterward, single-nuclei transcriptome information was loaded into RStudio using Seurat. Datasets corresponding to the primary plasmodium were merged, and nuclei with less than 3000 detected counts were used for further analyses using the Seurat package. The same process was accomplished for the datasets corresponding to the secondary plasmodium. The datasets were log normalized using the Seurat function (scale factor of 10,000). Neither feature selection nor mitochondrial gene removal was performed, and the data was directly standardized by regressing out differences in UMI and gene detection counts (Seurat package). Technical variation between runs within the same experiment was identified to be negligible. The dimensionality reduction process consisted of a PCA and UMAP with the first 15 PCs. Louvain clustering (Seurat default) was performed for the top 15 PCs with a resolution of 0.3. For the primary plasmodium, we identified cluster 3 as purely driven by low nUMI and nGene counts, and we therefore removed this cluster from the dataset. Afterward we re-clustered and re-embedded the dataset using only the top 7 PCs and a resolution of 0.4. All steps mentioned above were performed using the functions from the Seurat package. Differential gene expression analysis was performed using the Wilcoxon Rank Sum test (Seurat default) on identified clusters. Transcripts and cluster markers were queried to the UniProt annotation file obtained from *Glöckner and Marwan, 2017* and are listed in *Supplementary file 7*. The same procedure was applied to the Seurat objects containing all transcripts, and cluster information was transferred from the 'only annotated' objects, respectively, to allow a comparison of marker genes identified (*Supplementary file 7* and *Figure 3—figure supplement 1F and G*).

Cycling nuclei were extracted by cluster (1 and 2), and only marker genes identified for the three clusters of the spatial transcriptomic dataset of SM1 were used as input for PCA. Afterward the top 5 PC components were used for UMAP embedding and Louvain clustering. Pseudotemporal ordering of the cycling nuclei was performed using a diffusion map algorithm of the R package destiny (*Haghverdi et al., 2016*). Raw ordering results were separately obtained depending on the UMAP location ([c4],[c0/c2/c3],[c1]) and afterward sorted and ranked. Gene expression is visualized as a smoothed LOESS function (span = 10) against the pseudotemporal ordering.

Pseudobulk estimates were obtained for each cluster by averaging each gene's normalized expression value across the nuclei of the respective clusters. The same was done for the grid-based RNA-seq clusters of SM4. The pseudobulk matrix of the nuclei data was afterward correlated against the pseudobulk matrix of the grid-based spatial data using Spearman's correlation including only the top 50 cluster marker genes identified in the spatial RNA-seq dataset of SM4. Correlation values were scaled first by column and then by row, and values were cut to 1 or –1 when being either larger or smaller, respectively. Correlation values were used to hierarchical cluster the nuclei clusters using

the Ward method after calculating distances with Pearson's correlation. Afterward we selected two clusters (c0 and c4) and correlated the nuclei pseudobulk data against individual grids of SM4 using Spearman's correlation and the same gene set as used above. Raw correlation values were visualized as features on the grids and can be interpreted as either fan or network scores.

### GO term analysis

GO terms associated with cluster-specific marker genes (see *Supplementary file 7*) for SM1−4 and the primary and secondary plasmodium, respectively, were subjected to the R package GOseq (*Young et al., 2010*) to identify significantly (p<0.05) enriched GO terms per cluster. The enriched terms belonging to the category 'biological process' (BP) were loaded into the web browser tool Revigo (*Supek et al., 2011*) and were visualized with automatically lumping similar terms per plasmodium (*Figure 2—figure supplement 3*, *Figure 3—figure supplement 1H and I*).

### Analysis of scRNA-seq amoeba data

Again, only transcripts with a UniProt annotation or at least a gene description in the UniProt annotation file by *Glöckner and Marwan, 2017* were kept for further analyses while removing all other genes from the cell/gene matrices. There were 13,802 of the predicted 28,139 transcripts/genes in the transcriptome left for analysis. When unannotated transcripts are not removed, 22,905 transcripts remained for the analysis. The following analysis descriptions focus on the 'only annotated transcript' objects if not stated differently. Datasets of the two sequencing lanes were merged using Seurat, and cells with detected UMI counts ranging from 6000 to 25,000 were extracted for downstream analyses. Normalization and standardization steps were performed identically as described above. Neither feature selection nor mitochondrial gene removal was done. PCA was followed by UMAP and Louvain clustering with the first 25 PCs and a resolution of 0.6. Differential gene expression analysis was performed using the Wilcoxon Rank Sum test (Seurat default) on identified clusters (*Supplementary file 7*). The same procedure was applied to the Seurat object containing all transcripts, and cluster information was transferred from the 'only annotated' object to allow a comparison of marker genes identified (*Supplementary file 7* and *Figure 4—figure supplement 1*).

Cycling amoebae were extracted by cluster identity (cluster 3) and afterward merged with the Seurat object containing the spatially resolved transcriptomes of SM1. Data integration was performed using Seurat's built-in function for 'anchoring' data of different experiments to remove the effect of single-cell platforms used during data generation (*Stuart et al., 2019*). We used all genes for finding anchors between the datasets and afterward 20 PCs for the integration with the identified anchors. The UMAP embedding and Louvain clustering were then performed on the integrated data using the top 10 PCs and with a resolution of 0.4 for clustering.

## Acknowledgements

We thank all lab members of the Treutlein, Alim, and Camp labs for helpful discussions and feedback. In particular, we thank Damian Wollny for comments on the RNA extraction procedure. We are grateful for the help of Noah Ziethen and Björn Kscheschinski with sample collection and for the advice of Christian Westendorf on microscopy. We thank Antje Weihmann and Barbara Schellbach for performing sequencing runs and the Single Cell Facility (SCF) at D-BSSE for insights in dyes and imaging approaches. We are thankful for the interesting discussions with Oskar Camp that helped to initiate this research project. JGC and BT were supported by grant number CZF2019-002440 from the Chan Zuckerberg Initiative DAF, an advised fund of the Silicon Valley Community Foundation. JGC was supported by the European Research Council (Anthropoid-803441) and the Swiss National Science Foundation (Project Grant-310030_84795). BT was supported by the European Research Council (Organomics-758877, Braintime-874606), the Swiss National Science Foundation (Project Grant-310030_192604), and the National Center of Competence in Research Molecular Systems Engineering. TG, NS, SC, and KA received funding from the Max Planck Society.

## Additional information

### Funding

| Funder | Grant reference number | Author |
|---|---|---|
| Chan Zuckerberg Initiative | CZF2019-002440 | Barbara Treutlein |
| H2020 European Research Council | Anthropoid-803441 | J Gray Camp |
| H2020 European Research Council | Organomics-758877 | Barbara Treutlein |
| H2020 European Research Council | Braintime-874606 | Barbara Treutlein |
| Swiss National Science Foundation | Project Grant-310030_192604 | Barbara Treutlein |
| Swiss National Science Foundation | Project Grant-310030_84795 | J Gray Camp |
| Max Planck Institute for Dynamics and Self Organization | | Nico Schramma Siyu Chen Karen Alim |
| Max Planck Institute for Evolutionary Anthropology | | Tobias Gerber |

The funders had no role in study design, data collection and interpretation, or the decision to submit the work for publication.

### Author contributions

Tobias Gerber, Conceptualization, Data curation, Formal analysis, Investigation, Methodology, Visualization, Writing – original draft, Writing – review and editing; Cristina Loureiro, Formal analysis, Investigation, Methodology, Writing – original draft, Writing – review and editing; Nico Schramma, Formal analysis, Investigation, Methodology, Visualization, Writing – original draft, Writing – review and editing; Siyu Chen, Akanksha Jain, Investigation, Methodology; Anne Weber, Resources; Anne Weigert, Malgorzata Santel, Investigation; Karen Alim, Conceptualization, Funding acquisition, Project administration, Supervision, Writing – original draft, Writing – review and editing; Barbara Treutlein, J Gray Camp, Conceptualization, Funding acquisition, Project administration, Supervision, Visualization, Writing – original draft, Writing – review and editing

### Author ORCIDs

Tobias Gerber http://orcid.org/0000-0001-8456-5495
Karen Alim http://orcid.org/0000-0002-2527-5831
Barbara Treutlein http://orcid.org/0000-0002-3299-5597
J Gray Camp http://orcid.org/0000-0003-3295-1225

### Decision letter and Author response

Decision letter https://doi.org/10.7554/eLife.69745.sa1
Author response https://doi.org/10.7554/eLife.69745.sa2

## Additional files

### Supplementary files

• Supplementary file 1. Sample overview.

• Supplementary file 2. Metadata for spatial transcriptomics dataset SM1. Excel sheets containing sampling information, sequencing library pooling scheme, and metadata such as grid coordinates and sequencing depth per grid. Sequencing depth is provided as the number of paired reads per grid, and only the grids that are analyzed in the article are listed on sheet 3.

• Supplementary file 3. Metadata for spatial transcriptomics dataset SM2. Excel sheets containing sampling information, sequencing library pooling scheme, and metadata such as grid coordinates and sequencing depth per grid. Sequencing depth is provided as the number of paired reads per

grid, and only the grids that are analyzed in the article are listed on sheet 3.

• Supplementary file 4. Metadata for spatial transcriptomics dataset SM3. Excel sheets containing sampling information, sequencing library pooling scheme, and metadata such as grid coordinates and sequencing depth per grid. Sequencing depth is provided as the number of paired reads per grid, and only the grids that are analyzed in the article are listed on sheet 3.

• Supplementary file 5. Metadata for spatial transcriptomics dataset SM4. Excel sheets containing sampling information, sequencing library pooling scheme, and metadata such as grid coordinates and sequencing depth per grid. Sequencing depth is provided as the number of paired reads per grid, and only the grids that are analyzed in the article are listed on sheet 3.

• Supplementary file 6. Metadata for spatial transcriptomics dataset T1–3. Excel sheets containing sampling information, sequencing library pooling scheme, and metadata such as grid coordinates and sequencing depth per grid. Sequencing depth is provided as the number of total reads per grid.

• Supplementary file 7. Cluster marker and Gene Ontology (GO) term enrichments. Excel sheets containing cluster marker information and GOseq enrichments to explore the complete list of gene expression differences identified. Lists presented are sorted by the corresponding figure numbers.

• Transparent reporting form

## Data availability
Sequencing data have been deposited in GEO under accession code GSE173809.

The following dataset was generated:

| Author(s) | Year | Dataset title | Dataset URL | Database and Identifier |
|---|---|---|---|---|
| Treutlein, et al | 2022 | Spatial transcriptomic and single-nucleus analysis reveals heterogeneity in a gigantic single-celled syncytium | http://www.ncbi.nlm.nih.gov/geo/query/acc.cgi?acc=GSE173809 | NCBI Gene Expression Omnibus, GSE173809 |

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
