## [Editor Report]

Single-celled organisms are assumed to be smaller, simpler, and less complex than multicellular organisms like animals. Here, the authors provide evidence for variation in gene expression in the syncytial (multinucleate) large amoeba *Physarum polycephalum*. This study is an elegant and interesting regarding heterogeneity of gene expression patterns and thus specialization of functions within a syncytial organism.

---

## [Decision Letter]

**Decision letter after peer review:**

Thank you for submitting your article "Nuclei are mobile processors enabling specialization in a gigantic single- celled syncytium" for consideration by *eLife*. Your article has been reviewed by 3 peer reviewers, and the evaluation has been overseen by a Reviewing Editor and Marianne Bronner as the Senior Editor. The reviewers have opted to remain anonymous.

The reviewers have discussed their reviews with one another, and the Reviewing Editor has drafted this to help you prepare a revised submission. There is great enthusiasm about the problem and the system but also substantial technical concerns and over-interpretation of the data.

Essential revisions:

1) Data analysis: the existing data sets need to be more clearly explained in terms of controls, repetitions, and more explicitly discussing results. There are specific suggestions for data analysis that are raised in the reviews and each of these must be addressed.

2) Rationalize why certain genes were "cherry-picked" and use the sequencing data to provide a more thorough analysis and description of different classes of nuclear states.

3) Rewrite discussion so as to be less speculative and over-interpretative and instead directly address the main findings and implications. Suggestions for improving this are found in the specific comments below.

*Reviewer #1 (Recommendations for the authors):*

While primarily descriptive work, the authors are pushing a more provocative mechanistic interpretation of the scRNAseq results which is not supported in the current form of the manuscript. Other claims – such as the differential gene expression between nuclei within a syncytium are noteworthy and supported given the caveats below:

1. Lack of clarification of sampling regime and reps and controls

Throughout it is hard to determine how many replicate samples or conditions or type of samples (vein network, primary vein, growth front or fan – Figure 1B circles both network and fan region for example),. The introduction mentions "multiple reps" and conditions – apparent four total syncytia although really one biological sample for SM3 vs. SM4. Were all transcriptomes compared by all? A more detailed outline or table of samples and conditions (and rationales for DGE and data presentations) would help with interpretation of the DGE results and conditions compared.

What are cutoffs and why are different genes presented for DGE used for heat maps (Figures 2 -4)?

How were these marker genes chosen to be represented in spatial maps? The reasoning for the choice of these are unclear as is why they are different between samples. Using the same markers across replicates, and mor measuring different timepoints of chemotaxis to the oat engulfment would have been a clearer picture of what was happening transcriptionally in that experiment.

2. Issues with both interpreting and discussing differential gene expression (DGE)

DGE is always comparing one condition to another condition and thus is a comparison of datasets. "Up-" or "down-regulated" is thus relative and contextual. Throughout the manuscript (in heat maps and discussions) it remains unclear what is being compared that results in "up" v. "down" regulation. Thus Figures (1-4), Figure legends, Results and interpretations in the discussion need to reflect the direction of these comparisons and controls, otherwise it is both confusing and or misleading.

For 2B – clearly all samples are different, yet isn't what is being compared (or interesting) is how the regions are same/different not only within syncytia but between samples (not just SM4)? Indeed, "polycephalum" means literally multiple "heads" and thus comparisons between syncytial regions (even just the "heads") between and within samples (SM1-4) would provide a better understanding for how reproducible the scRNAseq results are.

What is "pseudotime" ordering? What are the criteria for this and why used? I think this could be dropped in lieu of actual data presentation that defines mitotic profiles and correlations with transcriptional regions with those profiles.

3. Conflation of Correlation vs. causation?

The primary claim of the manuscript is that "nuclei are 'mobile' processors facilitating specialized functions" see introduction and in Discussion "dynamic structure formation and response to local environmental conditions is coordinated in part through local gene regulation". The contention is that nuclei "adapt" to local and temporal stimuli and that transcriptional program is maintained long enough to be quantified. How long to nuclei stay within any given region (L398)? What is the cytological evidence for this?

This implies that transcriptional changes in nuclei facilitate developmental or specialized functions (fanning) of the syncytium. Yet given snapshot evidence differential gene expression within the syncytia – is this DGE a "cause" or "consequence" of spatial separation or other environmental cues such as those proposed for SM3? Clearly the oat grain results in DGE – likely sensory or chemotactic response. Thus, DGE could associated with growth front vs. network spatially and/or signal transduction/chemotaxis/or behavior could have multiple and complex causes and consequences. Concluding that the transcriptome state "associates" with behavior or location preserves rather than rectifies this ambiguity (e.g., lines 313-314) as the "behavior" could be defining the morphology (or vice versa). Even physically translocating a region of the syncytium in proximity to another region or modifying inputs could within the same syncytium could help with interpreting transcriptional changes.

First – the spatial separation resulting from the growing syncytium and reportioning of nuclei *causes* a transcriptional response within different nuclei. Alternatively, the transcriptional responses within different nuclei *causes* different developmental responses within the syncytium resulting in its differentiation. Or alternatively, the changes in length of the network and changes in the surface area of the fan region that result during growth responses also cause in transcriptional changes in syncytial nuclei. The snap-shot approaches used here do not allow the authors to distinguish between the alternative explanations, which ultimately would require some sort of temporal scanning combined with spatial scanning of transcriptional responses.

4. Sole inferences of cellular function based only on differences in gene expression.

Based on the upregulation of genes associated with SM1, the authors claim nuclei are actively dividing. This could easily be confirmed in the literature or by alternative methods that quantify protein expression or localization (immunostaining, Western blot etc.). Are cell cycle genes mitotic or G2/M phase markers transcriptionally upregulated? Further the conclusions regarding tubulin gene regulation are also a bit overstated as gene expression does not necessarily reflect protein levels, pools, or localization. Thus, while increased cell division based on DGE alone is consistent with the conclusions provided, these results along are not conclusive. Thus, the use of alternative approaches to vet the claims of mitosis and cytokinesis would greatly strengthen the manuscript and these conclusions.

5. Issues of heterogeneity in single amoebae to the variation observed in syncytia (Figure 4).

Why is this the case? How does this heterogeneity get homogenized in syncytia? It seems that the apt experiment would be to look at the developmental transition between single amoebae and the syncytium – otherwise this result kind of sticks out without much context to the rest of the manuscript or the mechanisms proposed as seen in the lack of clarification in lines 374-376.

6. Unsupported evolutionary arguments in Discussion

Comparisons are probably more apt to other syncytial cell types in multicellular organisms or in other Amoebozoa (L426). The discussion of evolution of multicellularity or viruses in the eukaryotic common ancestor are highly speculative and seem out of context. Syncytial types could be derived from tissue compartmentalization rather than a progenitor of such. These are intriguing points but come off as non sequiturs compare to the principles of spatially segregated (and likely temporally) segregated transcription within nuclei.

*Reviewer #2 (Recommendations for the authors):*

Review of "Nuclei are mobile processors enabling specialization in a gigantic single- celled syncytium". Physarum polycephalum is a member of Myxogastria, within the Amoebozoa, which is a sister clade to Opisthokonta, which includes animals and fungi. P. polycephalum grows as a multinucleate syncytial plasmodium, while other members of Amoebozoa grow as single nucleate cells (*Dictyostelium*). In this study, the authors use a clever method to isolate sections of a diploid plasmodium of P. polycephalum by growing a single plasmodium over a 384 well grid and isolating RNA from each colonized grids by centrifuging the 384 well plate. A total of 4 genetically identical plasmodia were assayed, two of which were grown in a uniform environment and two of which were exposed to oat flakes. The authors show variability in gene expression patterns across a plasmodium, with clustering of gene expression patterns in certain areas of the plasmodium, for example the growth front (fan) versus the vein network. Additionally, each of the four plasmodia also showed a unique expression patterns. Single nuclear RNAseq was also performed to assess expression patterns differences between nuclei within the syncytium and again, clustering of expression profiles was observed. Expression patterns in syncytial plasmodia was also compared to expression profiles obtained from single celled haploid amoeba. As between plasmodia, expression patterns differences were observed between individual amoebae, particularly in cell cycle functions, in addition to the expression of amoebae specific genes. This study is quite elegant and the data supports heterogeneity of gene expression patterns and thus specialization within a syncytial multinucleate plasmodium.

Some information regarding the genome and predicted genes in the P. polycephalum is needed in the introduction, strengths and deficiencies as a model and perhaps a discussion on previous work on this organism (mazes comes to mind). This would be useful for those readers not familiar with this organism. In particular, for the plasmodium section, it is unclear how many unique transcripts were detected across the entire 4 plasmodia and also within each cluster. The expression of how many of the predicted genes in the genome (~48,000) were detected in the clusters? The authors indicate that transcripts without predicted functional annotation were removed from analyses. "Only transcripts with a UniProt annotation or at least a gene description in the UniProt annotation file were kept for further analyses while removing all other genes from the grid/gene matrices."

How many were left for analyses? How was functional prediction of this group evaluated? Did any of these throw away genes show clustering with genes of predicted function? Some information on this aspect is needed.

The functions of different parts of the plasmodium were often extrapolated based on expression of marker genes whose function has been presumably characterized in other organisms. This aspect should be made clear-what is known is P. polycephalum and what is being extrapolated from distant relatives (humans, yeast, *Dictyostelium*). How were marker genes selected? How sure are the authors that these marker genes actually encode orthologs? What is the justification for selecting these particular genes to highlight, other than their potentially predicted role based on studies in other organisms. Similarity/conservation? Within an expression cluster, were particular functional categories enriched (such as cell cycle)? If so, add statistics to the manuscript text. The extrapolation of plasmodial function based on expression of a few marker genes without statistical support is speculative and is an issue throughout the manuscript.

Section starting at Line 316: Is there any enrichment for any functional categories of the single cell data? This section picks out particular transcripts without justification or statistical support other than their predicted function in other organisms.

In Videos 1 and 2, the nuclei show very different sizes and morphologies. Meaning??

Why is the number of reads from the single nucleus RNAseq (on average 591 transcripts (UMIs) per nucleus (Figure S4B)) in the plasmodium so much lower than single cell amoeba (14,525 transcripts per cell (Figure 4C))? Could this be biologically relevant? Does this invoke technical aspects associated with comparisons?

Within the plasmodium, using single nucleus RNAseq, did the number of reads per nucleus vary in the plasmodium? Could some nuclei be quiescent and transcriptionally inactive (for example, those that are moving) versus others that are anchored, as in Video 1?

For the supplemental datasets, the first two sheets are informative, and the total number of reads is ok (third sheet), but information regarding genes (particularly those with annotation, but also without annotation, but with gene ID) and expression levels within the various clusters in the plasmodium would be much more informative. As it is, these spreadsheets do not add a whole lot of understanding of the data generated in this study. It would be useful to have detailed information about the expression levels of genes in each cluster, along with the UniProt prediction and gene ID in Physarum. Otherwise, readers will have to re-analyze the deposited RNAseq data to derive this information and to verify the results presented in the article.

The discussion needs more evaluation of methods and results of data presented, for example, what does it mean that each plasmodium (in an identical environmental condition) shows dramatic differences in expression patterns, including different areas of the plasmodium? Two of the slime molds (SM1 And SM2) were in identical conditions and yet there was little overlap in expression patterns. A similar observation was also reported for the amoebae results. Some interpretation and discussion are needed.

In the discussion and data in Figure S4F is highlighted indicating "plasmodial specific expression of virus-related genes (RAP = Retroviral aspartyl protease domain)" and in the discussion. Was there enrichment for expression of virus-related genes? A discussion of these data should be included in the results and not the Discussion section. If this is the only "virus-related" gene and its function is unknown in Physarum, the inclusion of the discussion that it might play a role in syncytial formation is highly speculative.

*Reviewer #3 (Recommendations for the authors):*

I think the work stands for itself and could be published without major changes to the data for the most part. But really, the biggest improvement to the work seems that it would be to try to focus the presentation on more exciting biology uncovered by the data set. While the cell cycle is an easy signal to find, it doesn't seem to all that exciting? And in the end, I didn't understand how to put the pieces together: how do these different nuclear programs relate to the spatial gene expression and different nuclear behavior? In the end, I was left with the impression that this was mostly of interest to Physarum researchers, rather than a broader audience, given that it was not clear that it had revealed new biological principles.

[Editors’ note: further revisions were suggested prior to acceptance, as described below.]

Thank you for submitting your article "Spatial transcriptomic and single-nucleus analysis reveals heterogeneity in a gigantic single-celled syncytium" for consideration by *eLife*. Your article has been reviewed by 2 peer reviewers, and the evaluation has been overseen by a Reviewing Editor and Marianne Bronner as the Senior Editor. The reviewers have opted to remain anonymous.

The reviewers have discussed their reviews with one another, and the Reviewing Editor has drafted this to help you prepare a revised submission. The manuscript is much improved but there are still some outstanding points that need to be addressed by further revisions to the text. In particular, some important points that are presented as explanations in the rebuttal that should be incorporated the text.

Essential revisions:

1. From Rev #2 – the apt and fair question "how many genes were left for the analyses after dropping non-annotated genes?" needs to be addressed in the text as described in the author's response:

"We hope that the reviewer understands that adding the unannotated transcripts as the reviewer is asking for would cause a rework of all panels, which is not possible in the current state of the paper. We, however, encourage the reviewer and the slime mold community to download the raw data in order to check the influence of unannotated transcripts or to assign annotations based on co-expression with annotated genes in specific clusters."

This point is not only *not* addressed – it's punted to the reviewer to figure it out. It's not really a reviewer's role to reanalyze the data, rather more to evaluate claims and the evidence used to support those claims. This is an important point, as such "unannotated transcripts" are just those that lack homologs in organisms for which GO was developed – and the choice to drop them doesn't really have a specific reason other than to reduce the analyses. At least some specific information added to the text could account for the overall "spirit" of this comment – which is that a lot of new biology could be lurking in proteins that lack homology-based annotation. This point should be clearer in the analyses and the Discussion.

2. In response to the comment that the "discussion needs more evaluation of methods and results of data presented" the non-substantiated answer in the rebuttal was provided that "we also think that there are real differences between the individual slime molds and have therefore opted to keep the original plots in the main figures". The comment was provided to ask the reviewers to explain their logic – not double down on their point of view. It's a fair comment to ask that logic be explained when to the reader this is unclear. This logic would still need to be incorporated into the discussion.

3. Further GO enrichment analyses include a lot of functional assumptions that might not translate from say yeast protein function to Physarum protein function for the same homolog. The big assumption is that functional mitotic or chromosome annotation of genes in the Amoebozoan Physarum (e.g, Chromosome segregation or Spindle Assembly) are actually specific to those processes in Physarum. At minimum the limitations of relying on this sole method to assess functions of transcripts should be addressed in the Discussion.

4. Inclusion of the explanatory points made in the rebuttal that either "close" a question or point to the fact that a key point remains "open" and rectification of alternative explanations can yet be supported by this study should be included in the final version.

---

## [Author Response]

Reviewer #1 (Recommendations for the authors):While primarily descriptive work, the authors are pushing a more provocative mechanistic interpretation of the scRNAseq results which is not supported in the current form of the manuscript. Other claims – such as the differential gene expression between nuclei within a syncytium are noteworthy and supported given the caveats below:1. Lack of clarification of sampling regime and reps and controlsThroughout it is hard to determine how many replicate samples or conditions or type of samples (vein network, primary vein, growth front or fan – Figure 1B circles both network and fan region for example),. The introduction mentions "multiple reps" and conditions – apparent four total syncytia although really one biological sample for SM3 vs. SM4. Were all transcriptomes compared by all? A more detailed outline or table of samples and conditions (and rationales for DGE and data presentations) would help with interpretation of the DGE results and conditions compared.

We thank the reviewer for this comment and added Supplementary File 1 providing an overview of the sample regime and the sample composition in order to clarify which stages and parts have been sampled and how many replicates exist. Furthermore, we rephrased parts of the main text in order to be more precise where needed or broader when we are referring to slime mold parts for the imaging data in line 109-111.

What are cutoffs and why are different genes presented for DGE used for heat maps (Figures 2 -4)?

In order to make it more clear and transparent about which genes were used for visualizing which DGE result, we added a new table (Supplementary File 7) that summarizes all the genes found to be specific to clusters identified across the different experiments. We selected genes for visualization in heatmaps based on statistical tests implemented through the Seurat package, and tried to highlight convincing and interesting markers within these DEG lists in the heatmap visualizations.

How were these marker genes chosen to be represented in spatial maps? The reasoning for the choice of these are unclear as is why they are different between samples. Using the same markers across replicates, and mor measuring different timepoints of chemotaxis to the oat engulfment would have been a clearer picture of what was happening transcriptionally in that experiment.

Thank you for bringing up this point. In this experiment, we were primarily interested in determining if there was intra-plasmodial transcriptome heterogeneity, rather than consistency across plasmodia. We therefore performed a DGE analysis on the clusters identified for each plasmodium individually in order to assess and understand the heterogeneity within each individual plasmodium (Figure 2C-D) which is independent of the DGE analysis we performed on the combined data set in Figure 2B. We agree that more experiments in the future will be fun to shed light on plasmodia gene regulatory networks controlling specific and reproducible behaviours such as chemotaxis.

2. Issues with both interpreting and discussing differential gene expression (DGE)DGE is always comparing one condition to another condition and thus is a comparison of datasets. "Up-" or "down-regulated" is thus relative and contextual. Throughout the manuscript (in heat maps and discussions) it remains unclear what is being compared that results in "up" v. "down" regulation. Thus Figures (1-4), Figure legends, Results and interpretations in the discussion need to reflect the direction of these comparisons and controls, otherwise it is both confusing and or misleading.

We thank the reviewer for pointing us to this imprecise description and we have updated the text and figures accordingly. Briefly, DGE was always performed on nuclei/cell/grid clusters by determining which genes are specifically upregulated in a certain cluster compared to other clusters. We therefore changed the term ‘differentially expressed genes’ to ‘cluster specific gene expression’ throughout the manuscript, and provided explanations in the methods starting at line 803 and following paragraphs. In addition, we added Supplementary File 7 to provide additional clarity of which markers were selected.

For 2B – clearly all samples are different, yet isn't what is being compared (or interesting) is how the regions are same/different not only within syncytia but between samples (not just SM4)? Indeed, "polycephalum" means literally multiple "heads" and thus comparisons between syncytial regions (even just the "heads") between and within samples (SM1-4) would provide a better understanding for how reproducible the scRNAseq results are.

We are thankful for providing such great input. In our previous manuscript, we indeed highlighted what was variable within an individual syncytia, as we also think this is a very interesting question. However, we agree with the reviewer in that it is also interesting what is consistent between samples. We now emphasize and show in our revised manuscript also similarities between the fan structures across plasmodia (Figure 2- Figure supp. 1H-I) and also between the biological replicates SM1 and SM2 (Figure 2- Figure supp. 1D). In addition, we now integrate the different plasmodia grids by two methods, respectively, (Figure 2- Figure supp. 1E) allowing to regress out the batch effects in single-cell data. The two methods gave contradictive results (Figure 2- Figure supp. 1F) and we also believe that the integration process likely masks actual biological differences between the plasmodia sampled. However, one of the integrated data sets allowed us to identify further examples of gene expression signatures shared between plasmodium samples (Figure 2- Figure supp. 1G). Taken together, there are many similarities between the structures and replicates tested revealing new insights into molecular biological processes in *Physarum,* but also highlighting the robustness of our approach.

What is "pseudotime" ordering? What are the criteria for this and why used? I think this could be dropped in lieu of actual data presentation that defines mitotic profiles and correlations with transcriptional regions with those profiles.

We thank the reviewer for this comment. Because states along a temporal trajectory (e.g. mitosis) are present in one sample and we are able to sample the continuum of states through our methods, we are able to order the nuclei/cells/grid based on a continuum of expression profiles. This computation-based ordering is often referred to as “pseudotime”. We used the ‘pseudotime’ ordering to align the nuclei data of the primary plasmodium without spatial resolution to the grid data of SM1 with spatial resolution to compare mitotic ‘pseudotime’. We feel that this comparison is valid and shows that one can assign the nuclei data to the grid data depending on the different mitotic progression states. We adjusted the text in line 281-287 to explain more detailed the rationale of our method choice.

3. Conflation of Correlation vs. causation?The primary claim of the manuscript is that "nuclei are 'mobile' processors facilitating specialized functions" see introduction and in Discussion "dynamic structure formation and response to local environmental conditions is coordinated in part through local gene regulation". The contention is that nuclei "adapt" to local and temporal stimuli and that transcriptional program is maintained long enough to be quantified. How long to nuclei stay within any given region (L398)? What is the cytological evidence for this?

We are thankful for bringing up this point. Based on feedback from reviewers we have adjusted the title of the manuscript to “Spatial transcriptomic and single-nucleus analysis reveals heterogeneity in a gigantic single-celled syncytium” to better reflect the results and interpretations in the manuscript. The major observations we present are that nuclei can move, that different parts of the syncytium express different genes, and that single-nuclei express different genes that correlate with different parts of the syncytium. To us this is super interesting, and supports the idea that nuclei can be mobile, information encoding processors that facilitate localized responses Extrapolations based on our videos suggest that nuclei can reside for at least a few minutes (line 434) in one location. Moreover, we also added a section in the discussion in line 439-463 where we discuss that the average dwelling time in the highly branched fan region is 4 times longer suggesting that nuclei reaching the fan region are on average longer in contact with growing front specific signaling events to either adapt or even ‘seed’ a gene expression response. Our data strongly suggests that the local dwelling time might be the driving force for local gene expression differences. It will be exciting to follow up on this research to understand how local transcriptomes are regulated.

This implies that transcriptional changes in nuclei facilitate developmental or specialized functions (fanning) of the syncytium. Yet given snapshot evidence differential gene expression within the syncytia – is this DGE a "cause" or "consequence" of spatial separation or other environmental cues such as those proposed for SM3? Clearly the oat grain results in DGE – likely sensory or chemotactic response. Thus, DGE could associated with growth front vs. network spatially and/or signal transduction/chemotaxis/or behavior could have multiple and complex causes and consequences. Concluding that the transcriptome state "associates" with behavior or location preserves rather than rectifies this ambiguity (e.g., lines 313-314) as the "behavior" could be defining the morphology (or vice versa).

Yes, this is a very interesting point. We are not currently able to address cause or consequence, however we think that based on the types of genes and GO categories that associate with the spatial and environmental cues, we think it is reasonable to propose that the transcriptional profiles facilitate specialized functions. We have tried to take the reviewers point into account and rephrased portions of the manuscript (e.g. line 173-176 and lines 331-335) to highlight that we can not yet definitively rectify this ambiguity between cause and consequence.

Even physically translocating a region of the syncytium in proximity to another region or modifying inputs could within the same syncytium could help with interpreting transcriptional changes.

This is a fantastic suggestion for future experiments and we believe that our manuscript highlights that Physarum could be an exciting model system for exploring how we syncytia can coordinate responses to dynamic environmental conditions. Unfortunately, we are at the moment not able to translocate tissue and analyze translocated nuclei afterwards. The translocation of plasmodium parts leads to an immediate fusion of the translocated and original structures, thereby, releasing translocated nuclei into the shuttle flow which makes these nuclei inaccessible for comparative RNA-seq readouts.

First – the spatial separation resulting from the growing syncytium and reportioning of nuclei causes a transcriptional response within different nuclei. Alternatively, the transcriptional responses within different nuclei causes different developmental responses within the syncytium resulting in its differentiation. Or alternatively, the changes in length of the network and changes in the surface area of the fan region that result during growth responses also cause in transcriptional changes in syncytial nuclei. The snap-shot approaches used here do not allow the authors to distinguish between the alternative explanations, which ultimately would require some sort of temporal scanning combined with spatial scanning of transcriptional responses.

We thank the reviewer for these critical points and addressed this clear uncertainty of correlation versus causation by toning down the claims we make regarding nuclei as mobile processors. We mark this claim now as hypothesis that needs to be evaluated further and also removed the phrase from the title. We believe that our data allows to speculate on this aspect as we see indirect evidence that nuclei could potentially seed transcriptomic states in other regions where they are shuttled to as now more explicitly discussed in a reworked paragraph (lines 443-467).

Furthermore, we are thankful for the excellent ideas for future experiments and agree that more research needs to be done to prove our hypothesis and to clarify what causes what.

4. Sole inferences of cellular function based only on differences in gene expression.Based on the upregulation of genes associated with SM1, the authors claim nuclei are actively dividing. This could easily be confirmed in the literature or by alternative methods that quantify protein expression or localization (immunostaining, Western blot etc.). Are cell cycle genes mitotic or G2/M phase markers transcriptionally upregulated? Further the conclusions regarding tubulin gene regulation are also a bit overstated as gene expression does not necessarily reflect protein levels, pools, or localization. Thus, while increased cell division based on DGE alone is consistent with the conclusions provided, these results along are not conclusive. Thus, the use of alternative approaches to vet the claims of mitosis and cytokinesis would greatly strengthen the manuscript and these conclusions.

We thank the reviewer for raising this point and we agree that more validations and experiments are needed to fully characterize and understand the molecular processes underlying the mitotic wave across the syncytium. We and others in the field have not yet been able to establish immunostainings for cell cycle regulators in Physarum. However, the particular genes that we identify as differentially expressed have been used in multiple other studies revealing variation in cell cycle. In addition we have strengthened our claims by performing a GO enrichment analysis (Figure 2- Figure supp. 3 and Figure 3- Figure supp. 1F-G) and the term ‘Chromosome Segregation’ is enriched in a separate cluster than the term ‘Spindle assembly’ highlighting the temporal progression differences across the plasmodium.

5. Issues of heterogeneity in single amoebae to the variation observed in syncytia (Figure 4).Why is this the case? How does this heterogeneity get homogenized in syncytia? It seems that the apt experiment would be to look at the developmental transition between single amoebae and the syncytium – otherwise this result kind of sticks out without much context to the rest of the manuscript or the mechanisms proposed as seen in the lack of clarification in lines 374-376.

Thank you for pointing out this point, and we realize the integration of the single amoebae and the syncytium was not fully smooth. Our primary goal for this manuscript was to explore nuclei heterogeneity in the slime mold syncytium, and we felt that a comparison to heterogeneity in free-living amoebae could help us better interpret syncytium data. We fully agree with the reviewer, it would definitely be interesting to reconstruct the transition from amoebae to syncytium, however we did not focus on this aspect. In our hands, for Phyasrum it is technically challenging to recapitulate these transition events and survey the transition steps using snRNA-seq. Hopefully, we can manage this reconstruction in a future manuscript. In our revised manuscript we have tried to better integrate and explain the rationale for the single amoebae experiments. In our revised manuscript, we studied in more detail the gene sets that are expressed in amoeba compared to the plasmodia and found that the plasmodia express many additional genes, and also that there are intriguing differences in cell cycle regulation. We agree that our work focuses on the outcome of the molecular switch from amoeboid cell divisions to nuclei divisions within the plasmodium rather than the switch itself. However, we also hope that we can attract with our work mitosis experts to the *Physarum* community focusing on the outstanding phenomenon of cell cycle switching (open vs closed mitosis) between the amoeba and plasmodium state in *Physarum* with state-of-the-art techniques.

6.Unsupported evolutionary arguments in DiscussionComparisons are probably more apt to other syncytial cell types in multicellular organisms or in other Amoebozoa (L426). The discussion of evolution of multicellularity or viruses in the eukaryotic common ancestor are highly speculative and seem out of context. Syncytial types could be derived from tissue compartmentalization rather than a progenitor of such. These are intriguing points but come off as non sequiturs compare to the principles of spatially segregated (and likely temporally) segregated transcription within nuclei.

We thank the reviewer for this critical point and agree that the data support for inferring evolutionary processes is highly speculative and we therefore decided to remove these paragraphs, sentences and figure panels from the manuscript.

Reviewer #2 (Recommendations for the authors):Review of "Nuclei are mobile processors enabling specialization in a gigantic single- celled syncytium". Physarum polycephalum is a member of Myxogastria, within the Amoebozoa, which is a sister clade to Opisthokonta, which includes animals and fungi. P. polycephalum grows as a multinucleate syncytial plasmodium, while other members of Amoebozoa grow as single nucleate cells (*Dictyostelium*). In this study, the authors use a clever method to isolate sections of a diploid plasmodium of P. polycephalum by growing a single plasmodium over a 384 well grid and isolating RNA from each colonized grids by centrifuging the 384 well plate. A total of 4 genetically identical plasmodia were assayed, two of which were grown in a uniform environment and two of which were exposed to oat flakes. The authors show variability in gene expression patterns across a plasmodium, with clustering of gene expression patterns in certain areas of the plasmodium, for example the growth front (fan) versus the vein network. Additionally, each of the four plasmodia also showed a unique expression patterns. Single nuclear RNAseq was also performed to assess expression patterns differences between nuclei within the syncytium and again, clustering of expression profiles was observed. Expression patterns in syncytial plasmodia was also compared to expression profiles obtained from single celled haploid amoeba. As between plasmodia, expression patterns differences were observed between individual amoebae, particularly in cell cycle functions, in addition to the expression of amoebae specific genes. This study is quite elegant and the data supports heterogeneity of gene expression patterns and thus specialization within a syncytial multinucleate plasmodium.

We are pleased to hear that the reviewer finds our experimental set up clever and the study elegant and we hope that we could adequately address all the points the reviewer raised in the following paragraphs.

Some information regarding the genome and predicted genes in the P. polycephalum is needed in the introduction, strengths and deficiencies as a model and perhaps a discussion on previous work on this organism (mazes comes to mind). This would be useful for those readers not familiar with this organism.

We thank the reviewer for this point and we added more examples which studies have been performed with *Physarum* to the introduction in lines 67-86 and mention the genome publication in line 90. In addition, we extended the introduction by providing more information on the nuclei division states in slime molds in lines 61-66. We hope to now strengthen the point that *Physarum* is an intriguing model organism to study syncytial dynamics, environmental response, mitotic mechanisms and how different nuclei states can be maintained while being exposed to the same cytoplasm.

We want to point out that the genome was not used for mapping (see point below) due to relatively low quality mapping statistics. Instead, we directly used the transcriptome for expression quantification, which has been slightly reworked in 2017 by Glockner and Marwan. This improved assembly and annotation greatly enhanced the interpretability of the data. We are happy to share more information on the mapping statistics if the reviewer is interested.

In particular, for the plasmodium section, it is unclear how many unique transcripts were detected across the entire 4 plasmodia and also within each cluster. The expression of how many of the predicted genes in the genome (~48,000) were detected in the clusters? The authors indicate that transcripts without predicted functional annotation were removed from analyses. "Only transcripts with a UniProt annotation or at least a gene description in the UniProt annotation file were kept for further analyses while removing all other genes from the grid/gene matrices."How many were left for analyses? How was functional prediction of this group evaluated? Did any of these throw away genes show clustering with genes of predicted function? Some information on this aspect is needed.

We thank the reviewer for bringing these gaps to our attention, and we agree that there is important information lacking regarding the transcriptome used and the gene filtering applied. We now indicate more clearly that we used the transcriptome for mapping in line 89 and we added to the methods section a more thorough description of the reference used. We state in line 808 that 15,197 of the predicted 28,139 transcripts in the transcriptome of Glockner and Marwan remained for the analysis after filtering for annotation and that we did not further explore aspects of the transcripts without annotations in the methods section. Furthermore, we added a panel in Figure 4D that shows that across all the samples we detect 14,785 transcripts of the 15,197 transcripts to be expressed at least in one of the sample types at a 10% quantile cutoff to exclude very lowly expressed genes. We hope that the reviewer understands that adding the unannotated transcripts as the reviewer is asking for would cause a rework of all panels, which is not possible in the current state of the paper. We, however, encourage the reviewer and the slime mold community to download the raw data in order to check the influence of unannotated transcripts or to assign annotations based on co-expression with annotated genes in specific clusters.

The functions of different parts of the plasmodium were often extrapolated based on expression of marker genes whose function has been presumably characterized in other organisms. This aspect should be made clear-what is known is P. polycephalum and what is being extrapolated from distant relatives (humans, yeast, *Dictyostelium*). How were marker genes selected? How sure are the authors that these marker genes actually encode orthologs? What is the justification for selecting these particular genes to highlight, other than their potentially predicted role based on studies in other organisms. Similarity/conservation? Within an expression cluster, were particular functional categories enriched (such as cell cycle)? If so, add statistics to the manuscript text. The extrapolation of plasmodial function based on expression of a few marker genes without statistical support is speculative and is an issue throughout the manuscript.

We thank the reviewer for raising this point, and we have tried to make it more clear in the revised manuscript what information was known from *P. polycephalum* and what is extrapolated from other species. There is very little known about protein function that has been learned in *P. polycephalum* and hence much of the interpretations come from orthology to other species, and we highlight this caveat in the revised manuscript. Whenever possible we tried to use genes annotated from *Dictyostelium* for which a wealth of molecular work has been collected over the last years while being a rather close relative to Physarum. We also performed a GO enrichment analysis to account for the lack of statistical testing and non-transparent gene selection (See also point below). We find a clear overlap of enriched GO terms and the inferred function of genes based on other organisms for the different slime molds as shown in multiple new panels (Figure 2- Figure supp. 3 and Figure 3- Figure supp. 1F-G). To be as transparent as possible we added Supplementary File 7 including all results for cluster specific genes and GO enrichments in order to allow for tracing back which genes and GO terms were selected and for easily exploring the data set for other genes of interest for the community.

Section starting at Line 316: Is there any enrichment for any functional categories of the single cell data? This section picks out particular transcripts without justification or statistical support other than their predicted function in other organisms.

We thank the reviewer for asking for functional categories which were indeed missing in the previous version of the manuscript. We performed a GO enrichment analysis for spatial transcriptomic samples SM1-4, respectively, and the plasmodia used for snRNA-seq experiments using the R package GOseq. The results were visualized by the R package Revigo which allows to lump highly similar GO term categories. The resulting dot plots of ‘biological processes’ are added as Figure 2- Figure supp. 3 and as panels in Figure 3- Figure supp. 1F-G. There are multiple interesting enrichments, and we find that there is a clear overlap of gene function inferred from other organisms and the slime mold specific GO enrichments.

In Videos 1 and 2, the nuclei show very different sizes and morphologies. Meaning??

We agree with the reviewer that there is an explanation of the different nuclei sizes needed and we added this information to the captions of the supplementary videos. The different sizes are predominantly a technical artefact which arise from the single imaging plane applied to the three-dimensional plasmodium tubes. However, we accounted for this technical issue (see Methods section) when imaging and segmenting nuclei in fixed plasmodia samples shown in Figure 1 in order to get a clean size distribution of nuclei. Nevertheless, there is some autofluorescence of presumably digestive vacuoles or other unknown aggregates occasionally visible in the shuttle flow (especially in Video 1) adding even more variation to the size distribution compared to fixed samples where no autofluorescence was detected.

Why is the number of reads from the single nucleus RNAseq (on average 591 transcripts (UMIs) per nucleus (Figure S4B)) in the plasmodium so much lower than single cell amoeba (14,525) transcripts per cell (Figure 4C)? Could this be biologically relevant? Does this invoke technical aspects associated with comparisons?

These are good questions and we are happy that we can clarify this point. We changed the term ‘nUMI’ to ‘nCounts’ in the figures accordingly and added a better explanation to the captions and the text. We used counts as a quantitative measure that describe the absolute transcript counts measured per nucleus/cell and are therefore reflecting the difference in absolute mRNA content between low mRNA content nuclei and the mRNA-rich whole-cell data. In contrast, we used nGene to describe the qualitative difference regarding the transcript diversity which is relatively as well as absolutely higher in the nuclei data. We are thankful for bringing up this point and added a new panel (Figure 4D) and discuss these findings in line 351-356.

Within the plasmodium, using single nucleus RNAseq, did the number of reads per nucleus vary in the plasmodium? Could some nuclei be quiescent and transcriptionally inactive (for example, those that are moving) versus others that are anchored, as in Video 1?

We thank the reviewer for this interesting point. It is a difficult question to answer. We did look at the distribution of reads per nucleus, and see a typical distribution similar to what we have observed in other snRNA-seq experiments. It would be difficult to distinguish between a poor quality nucleus from technical issues in the snRNA-seq protocol and something biologically interesting in vivo. In typical sn/scRNA-seq workflows, and in our data analyses, we filter out low expressing nuclei. As a consequence, we don’t see strong differences in transcript counts across the different clusters. A next step would be to find ways how to specifically label, track and isolate moving/stuck nuclei in order to understand if nuclei retain a memory of their current state or to evaluate the impact of dwelling time at a certain place in the plasmodium, or to explore this question more thoroughly using in situ imaging/sequencing. There is a lot of autofluorescence in the plasmodium which makes it difficult for image-based approaches*.* Unfortunately, we have not yet established the methods available that would allow yet to satisfactorily answer the reviewers question.

**Author response image 1. sa2fig1:** Transcript counts per nucleus for snRNA-seq data for the primary plasmodium (left) and the secondary plasmodium (right).

For the supplemental datasets, the first two sheets are informative, and the total number of reads is ok (third sheet), but information regarding genes (particularly those with annotation, but also without annotation, but with gene ID) and expression levels within the various clusters in the plasmodium would be much more informative. As it is, these spreadsheets do not add a whole lot of understanding of the data generated in this study. It would be useful to have detailed information about the expression levels of genes in each cluster, along with the UniProt prediction and gene ID in Physarum. Otherwise, readers will have to re-analyze the deposited RNAseq data to derive this information and to verify the results presented in the article.

We thank the reviewer for this point and agree that some information in the previous supplementary tables were only relevant when reanalyzing the raw data of our study. As mentioned earlier, we added a new Supplementary File 7 that contains marker genes with predicted annotations per cluster for all the embeddings shown in the figures of the manuscript. In addition, the table contains the GO enrichments for the clusters. We hope that the table allows an easy exploration of the data for other researchers.

The discussion needs more evaluation of methods and results of data presented, for example, what does it mean that each plasmodium (in an identical environmental condition) shows dramatic differences in expression patterns, including different areas of the plasmodium? Two of the slime molds (SM1 And SM2) were in identical conditions and yet there was little overlap in expression patterns. A similar observation was also reported for the amoebae results. Some interpretation and discussion are needed.

We are thankful for bringing this up, and we agree with the reviewer that it is difficult to distinguish between differences between slime molds and technical issues associated with batch. In our revised manuscript, we have also provide an integrated view of spatial heterogeneity across the slime molds (Figure 2- Figure supp. 1D-F) which integrates the datasets (by definition of the algorithm), and highlights the similarities between replicates and similar structures (e.g. fan regions across different slime molds). This analysis emphasizes that the data is not only explained by differences. However, we also think that there are real differences between the individual slime molds, and have therefore opted to keep the original plots in the main figures. We have provided more details in the methods section.

In the discussion and data in Figure S4F is highlighted indicating "plasmodial specific expression of virus-related genes (RAP = Retroviral aspartyl protease domain)" and in the discussion. Was there enrichment for expression of virus-related genes? A discussion of these data should be included in the results and not the Discussion section. If this is the only "virus-related" gene and its function is unknown in Physarum, the inclusion of the discussion that it might play a role in syncytial formation is highly speculative.

We thank the reviewer for this critical point and agree that the data support for inferring virus driven evolutionary processes is highly speculative. Based on comments from multiple reviewers, we therefore decided to remove this paragraph from the manuscript while still being intrigued by this idea.

[Editors' note: further revisions were suggested prior to acceptance, as described below.]

Essential revisions:1. From Rev #2 – the apt and fair question "how many genes were left for the analyses after dropping non-annotated genes?" needs to be addressed in the text as described in the author's response:"We hope that the reviewer understands that adding the unannotated transcripts as the reviewer is asking for would cause a rework of all panels, which is not possible in the current state of the paper. We, however, encourage the reviewer and the slime mold community to download the raw data in order to check the influence of unannotated transcripts or to assign annotations based on co-expression with annotated genes in specific clusters."This point is not only not addressed – it's punted to the reviewer to figure it out. It's not really a reviewer's role to reanalyze the data, rather more to evaluate claims and the evidence used to support those claims. This is an important point, as such "unannotated transcripts" are just those that lack homologs in organisms for which GO was developed – and the choice to drop them doesn't really have a specific reason other than to reduce the analyses. At least some specific information added to the text could account for the overall "spirit" of this comment – which is that a lot of new biology could be lurking in proteins that lack homology-based annotation. This point should be clearer in the analyses and the Discussion.

We fully agree on these points. We have added a new section to the Discussion that explains the limitations of our analyses, as well as the limitation to the current *Physarum* genome and transcriptome annotations. We are hoping that one outcome of our manuscript is to stimulate a renewed and vigorous interest in *Physarum* genomics. We do want to explain to the reviewers and editor that we made the decision to focus only on annotated transcripts because of the relatively poor quality of the *Physarum* genome and transcriptome. In addition, we felt that we could only interpret heterogeneity on the basis of annotated features. Nevertheless, we repeated a number of analyses including all transcripts and found that the input gene selection has no major impact on the overall clustering outcome. We discuss this at the respective positions in the revised manuscripts (Figure 2 —figure supplement 1H; Figure 3 —figure supplement 1F and G; Figure 4 —figure supplement 1A). We also provide all cluster marker information that includes unannotated features for the new panels as separate Excel sheets to Supplementary File 7.

2. In response to the comment that the "discussion needs more evaluation of methods and results of data presented" the non-substantiated answer in the rebuttal was provided that "we also think that there are real differences between the individual slime molds and have therefore opted to keep the original plots in the main figures". The comment was provided to ask the reviewers to explain their logic – not double down on their point of view. It's a fair comment to ask that logic be explained when to the reader this is unclear. This logic would still need to be incorporated into the discussion.

We fully agree on these points. We have added a new section to the Discussion that explains the limitations of our analyses, and explains our logic.

3. Further GO enrichment analyses include a lot of functional assumptions that might not translate from say yeast protein function to Physarum protein function for the same homolog. The big assumption is that functional mitotic or chromosome annotation of genes in the Amoebozoan Physarum (e.g, Chromosome segregation or Spindle Assembly) are actually specific to those processes in Physarum. At minimum the limitations of relying on this sole method to assess functions of transcripts should be addressed in the Discussion.

As mentioned above, we added a new paragraph explaining the limitations of our analyses including the interpretation of inferred transcript functions in *Physarum*.

4. Inclusion of the explanatory points made in the rebuttal that either "close" a question or point to the fact that a key point remains "open" and rectification of alternative explanations can yet be supported by this study should be included in the final version.

We went through the previous rebuttal and included the explanatory points into the manuscript at various positions. Briefly, we added the previous rebuttal figure on UMI count distributions across single-nuclei clusters to panel B in Figure 3—figure supplement 1 to visualize the homogenous distribution of transcript counts across nuclei subpopulations detected and describe this finding in line 285. We addressed all of the open points in the new Discussion sections.